# Tet2 Controls the Responses of β cells to Inflammation in Autoimmune Diabetes

Jinxiu Rui[1], Songyan Deng[1], Ana Luisa Perdigoto [1], Gerald Ponath[2], Romy Kursawe[3], Nathan Lawlor[3], Tomokazu Sumida [2], Maya Levine-Ritterman [2], Michael L. Stitzel [3,4], David Pitt[2], Jun Lu[5] & Kevan C. Herold [1✉]

β cells may participate and contribute to their own demise during Type 1 diabetes (T1D). Here we report a role of their expression of Tet2 in regulating immune killing. Tet2 is induced in murine and human β cells with inflammation but its expression is reduced in surviving β cells. Tet2-KO mice that receive WT bone marrow transplants develop insulitis but not diabetes and islet infiltrates do not eliminate β cells even though immune cells from the mice can transfer diabetes to NOD/*scid* recipients. Tet2-KO recipients are protected from transfer of disease by diabetogenic immune cells.Tet2-KO β cells show reduced expression of IFNγ-induced inflammatory genes that are needed to activate diabetogenic T cells. Here we show that Tet2 regulates pathologic interactions between β cells and immune cells and controls damaging inflammatory pathways. Our data suggests that eliminating TET2 in β cells may reduce activating pathologic immune cells and killing of β cells.

[1] Departments of Immunobiology and Internal Medicine, Yale University, New Haven, CT, USA. [2] Department of Neurology, Yale School of Medicine, New Haven, CT, USA. [3] The Jackson Laboratory for Genomic Medicine, Farmington, CT, USA. [4] Department of Genetics and Genome Sciences and Institute for Systems Genomics, University of Connecticut, Farmington, CT, USA. [5] Department of Genetics, Yale University, New Haven, CT, USA. ✉email: kevan.herold@yale.edu

Type 1 diabetes (T1D) is a chronic autoimmune condition that occurs over years after the first signs of autoimmunity, which are represented by the appearance of autoantibodies. With the improvement in the sensitivity of measurement of C-peptide and the availability of tissue sections from individuals who have died with T1D, it has become clear that not all β cells are destroyed by autoimmunity. Even 50 years after the diagnosis of T1D about 2/3 of patients still have detectable levels of C-peptide indicating residual β cells[1]. Moreover, the increased relative levels of proinsulin suggest dysfunctional but persistent β cells[2]. The reasons why some β cells survive and others succumb to autoimmune killing are not certain.

There is growing evidence showing β cells are not passive bystanders of their own destruction, but instead participate and contribute to their own demise in the pathogenesis of auto-immune diabetes. Genome-wide association studies (GWAS) show that >50% of gene loci associated with T1D are expressed in β cells. These studies, coupled with emerging molecular evidence that β cells are impaired by gain-of-function or loss-of-function of these loci, suggest an active role for the β cell in eliciting its own demise[3–6]. In addition to the intrinsic features of β cells, accumulating data suggest that β cells can control pathways of immune tolerance. β cells are required to initiate diabetes auto-immunity. Spleen cells from β-cell-deprived NOD mice cannot transfer diabetes although they maintain immune competency. Furthermore, "diabetogenic" spleen cells showed a reduced capacity to transfer diabetes after transient "parking" in β-cell-deprived mice[7]. The islet cell mass plays a critical role in diabetes —surgical removal of 90% of pancreatic tissue before the onset of insulitis induced a long-term protection in NOD mice. Lymphocytes from pancreatectomized diabetes-free mice exhibited a low response to islet cells[8]. Curiously, in NOD mice and human T1D, autoimmunity does not extend to cells that share antigens with β cells, particularly cohabitating endocrine islet cells and neuronal cells that express GAD and IA-2. It is thus likely that features of β cells that cause expression of autoantigens and their presentation under conditions for T cell activation are pre-requisites to the development of autoimmune lesions. Thus, the interplay between β cells and immune cells determine the outcome of autoimmunity.

β cells are also known to be intrinsically vulnerable to killing, and are more prone than other islet endocrine cells, to death under ER and immune stress conditions[9,10]. Early physiological β cell death triggers priming of autoreactive T cells by dendritic cells in pancreatic lymph nodes (pLNs)[11,12]. Thompson et al. described a subpopulation of β cells that become senescent and actively promotes the immune-mediated destruction process[13]. Recently Lee et al. reported that modulating the unfolded protein response (UPR) in NOD β cells by deleting the UPR sensor IRE1α prior to insulitis induced transient dedifferentiation of β cells, resulting in substantially reduced islet immune cell infiltration and β cell apoptosis and protection from diabetes[14]. When we previously analyzed β cells during the progression of diabetes, we identified a subset of β cells with dedifferentiated features such as reduced expression of Pdx1, Nkx6.1, MafA, and Ins1, Ins2 as well as β cell autoantigens, and those cells were resistant to immune-mediated killing[15].

Among the mechanisms that might account for the changes of β cells in response to stressors are epigenetic changes involving silencing or activation of genes as a consequence of signaling by factors such as inflammatory cytokines. Stefan-Lifshitz et al. associated DNA hydroxymethylation by IFNα[16]. In our previous studies, we found epigenetic changes in β cells from NOD mice that led to methylation marks in Ins1 and Ins2 and reduced gene transcription and similar changes in INS when human islets were cultured with inflammatory cytokines[17]. The increased expression of Dnmts, we postulated resulted in methylation of CpG sites in the insulin genes and repression of gene transcription[17].

The responses to inflammation and epigenetic modifications may be linked by the activity of the ten-eleven translocation (Tet) methylcytosine dioxygenase family members. Zhang et al. showed that Tet2 is required to resolve inflammation by recruiting Hdac2 to repress IL-6. In melanoma and colon tumor cells deletion of Tet2 reduced chemokine expression and tumor-infiltrating lymphocytes, enabling tumors to evade antitumor immunity and to resist anti-PD-L1 therapy[18]. Whereas enhancing TET activity with ascorbate/vitamin C increased chemokines, tumor-infiltrating lymphocytes, and antitumor immunity[18,19]. In primary microglial cells lacking Tet2, there is inhibition of IL-6 release and TNF-α after LPS treatment[20]. Deletion of Tet2 in T cells decreased their cytokine expression, associated with reduced p300 recruitment[21]. Lastly, a role for Tet2 in cell survival has also been suggested as Tet2-KO hematopoietic stem and progenitor cells have reduced apoptosis[22].

To understand how epigenetic modifications may change β cells and affect immune/β cell interactions during autoimmune diabetes, we studied the role of Tet2 and epigenetic changes in human and murine β cells during the autoimmune attack. Here we show that Tet2 regulates pathologic interactions between β cells and immune cells in humans with immune infiltrates in the pancreas and with diabetes and in mice with autoimmune diabetes by controlling damaging inflammatory pathways.

## Results

**Induction of Tet2 in murine β cells during the progression of autoimmune diabetes**. We first studied Tet gene expression in whole islets and β cells from NOD mice, a model of human T1D. Soon after weaning, at 3–4 weeks, a period of time that has been associated with the initiation of autoimmunity, we found a modest increase in expression of Tet1 but substantially increased expression of Tet2 in whole islets (Fig. 1a) in NOD mice but not immune-deficient NOD/scid mice of the same age. We identified these changes in Tet genes in β cells from 8- and 11-week-old NOD mice by enriching for β cells by sorting on Zinc + and TMRE + islet cells. (Fig. 1b). Because of the profound increase in Tet2 expression in NOD β cells, we focused our studies on this epigenetic regulator. We examined individual inflammatory cytokine as well as cytokine combinations which have been found to mediate β cell killing during the pathogenesis of autoimmune diabetes, on Tet2 induction in islets from B6 mice[23]. IL-1β or IFNγ alone and combinations of IL-6/IL-17A and TNFα/ IL-1β/ IFNγ could induce Tet2 gene transcription in the islets and β cells (Fig. 1c) as well as increased protein expression, identified by FACS (Fig. 1d).

**Expression of TET2 in human β cells**. We analyzed TET2 expression in human islets in different inflammatory settings by immunohistochemistry. In control pancreases, TET2 expression was seen outside of the islets among exocrine tissue (Fig. 2a(a)). However, there was increased TET2 expression in the nuclear and cytoplasm in β cells from a patient with autoimmune pancreatitis (with cellular immune infiltrates) (Fig. 2a(b)), as well as a patient with T1D autoantibodies but not diagnosed with T1D (Fig. 2a(c)), and patients with recent-onset T1D (Fig. 2a(d)). The expression of TET2 in β cells was associated with inflammatory lesions since we did not identify it in the β cells in the pancreas from patients with chronic pancreatitis in which infiltrating immune cells were not found (Fig. 2a(f)). Interestingly, we also did not identify TET2 staining in the remaining singular β cells in the pancreas from patients with long-established T1D who did not have detectable autoantibodies or insulitis (Fig. 2a(e)). The

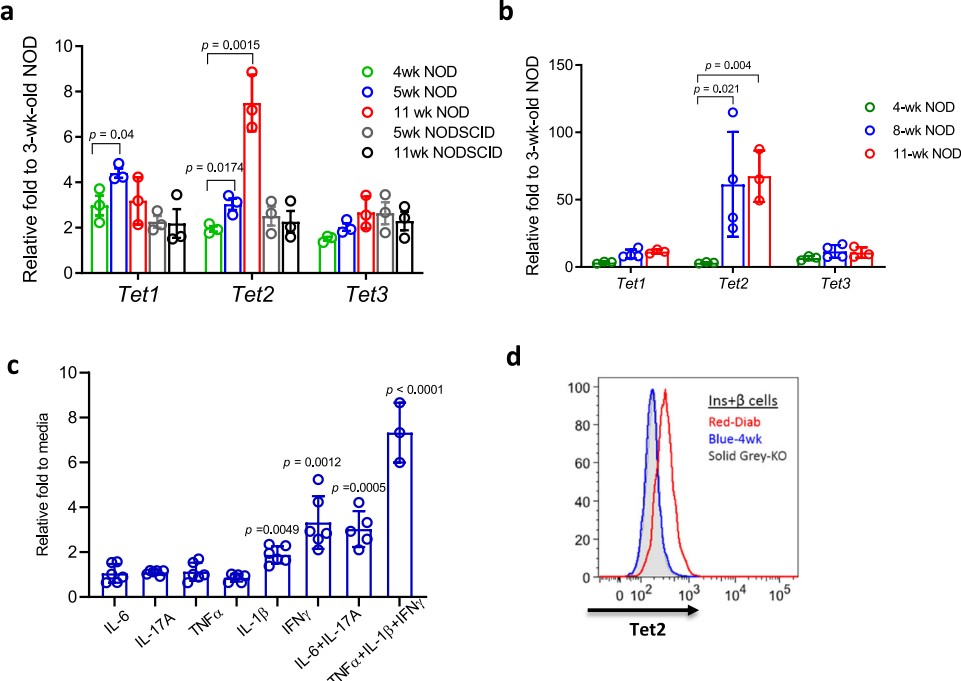

**Fig. 1 Increased *Tet2* expression in islet β cells during diabetes progression in nonobese diabetic (NOD) mice.** Transcription analysis of the *Tet* genes in **a** handpicked islets from NOD and NOD/*scid* mice of different ages, **b** enriched β cells (Zn + TMRE+) sorted from NOD mice of different ages, and **c** islets from B6 mice following 24-h culture with indicated cytokines, 10 ng/ml each. RNA was recovered and *Tet* genes were measured by qRT-PCR. The Ct values were normalized to *Actb* mRNA levels (Delta Ct = (Ct of *Actb* − Ct of target gene) + 20). Data show the mean ± SD of three experiments, each with 3–6 mice. Statistical analysis was performed using two-way ANOVA with Tukey's multiple comparisons test in **a**, **b** and one-way ANOVA with Dunnett's multiple comparisons test in **c**. **d** Histogram comparing Tet2 protein level in β cells from 4-week-old as well as new-onset diabetic NOD mice analyzed by FACS. β cells from NOD Tet2-knock out (KO) mice were included as a negative control for Tet2 staining. Tet2+ β cells were identified by intracellular staining with antibodies against insulin and Tet2. Data were representative of three experiments, n = 3–5 for each experiment. (Gray histogram = β cells from NOD KO mice, Blue = wild-type NOD mice at 4 weeks of age and red = at diagnosis with hyperglycemia).

fluorescence intensity of *TET2* staining in β cells in different clinical conditions was quantified in Fig. 2b. The expression of *TET2* in β cells in autoimmune pancreatitis and T1D onset plus some autoantibody + individuals was significantly higher compared to conditions without inflammation. Our findings were consistent with our studies with murine islets and suggested that inflammatory cytokines could induce *TET2* expression. We, therefore, analyzed *TET2* expression in human islets (Supplementary Table 1) that were cultured with individual cytokine or cytokine combinations and found increased *TET2* gene expression in IL-1β or IFNγ alone as well as IL6/IL-17A and TNFα/IL-1β/IFNγ combinations group (Fig. 2c).

These observations suggest that *TET2* expression is induced by inflammatory cytokines and suggest a permissive but not sufficient role for β cell killing since we found increased *TET2* in β cells from patients with autoimmune pancreatitis that do not develop diabetes. However, we noted that not all of the β cells in patients with T1D or in the autoantibody + relatives were TET2+. Moreover, in samples from patients with long-standing T1D, we identified β cells by insulin staining, consistent with reports of residual β cells even in long-standing patients, but we did not find *TET2* expression in those β cells (Fig. 2a(e)).

**Resistance of Tet2-KO β cells to immune killing.** To understand the role of Tet2 in the inflammatory responses of β cells, we bred Tet2-KO NOD mice. Tet proteins are known to play an important role in cell development[24] and Dhawan et al. reported that deletion of *Tet2* in the pancreatic lineage results in improved glucose tolerance and β cell function (https://doi.org/10.2337/db18-50-OR). Therefore, we first characterized β cell function,

islet cells, and gene expression from Tet2-KO B6 mice. The glucose tolerance to IPGTTs and morphology of the islets were indistinguishable between the KO and wild-type (WT) mice (Supplementary Fig. 1a, b). There were small but significant differences in the proportion of β and α cells in the KO mice as well as in the MFI of insulin and Pdx1 but the expression of *Ins1, Ins2, Nkx6.1, ChgA,* and *Pdx1* were similar to WT β cells (Supplementary Fig 1c–e). Overall these findings suggest that there may be subtle differences in the composition of islets from KO and WT mice but not detectable functional abnormalities of Tet2-KO β cells.

We backcrossed the KO allele to NOD for more than 14 generations. The Tet2-KO NOD mice did not develop spontaneous diabetes whereas the median time to diabetes was 17 and 23 weeks in their WT and HET littermates (Supplementary Fig. 2).

Because Tet2 is involved in the development and function of immune cells, we transplanted bone marrow (BM) from WT NOD mice into WT or Tet2-KO recipients to eliminate any impact of Tet2 loss in immune cells. The Tet2-KO mice had a markedly reduced incidence of diabetes (median survival undefined) compared to WT recipients of NOD WT BM (median survival = 11 weeks) (p < 0.0001) (Fig. 3a). This was not due to the lack of immune infiltration in KO recipients. Figure 3b shows the frequencies of immune and nonimmune (CD45+ and −, respectively) from BMT recipients ages 8 to 20 weeks. We assessed the insulitis from KO BMT recipients at 20 weeks and analyzed 112 islets using a 4 point score (0–3). Forty-eight islets are score 0 (43%), 25 are score1 (22%), 26 are score 2 (23%), and 13 are score 3 (17%). At the same level of immune cell infiltrates,

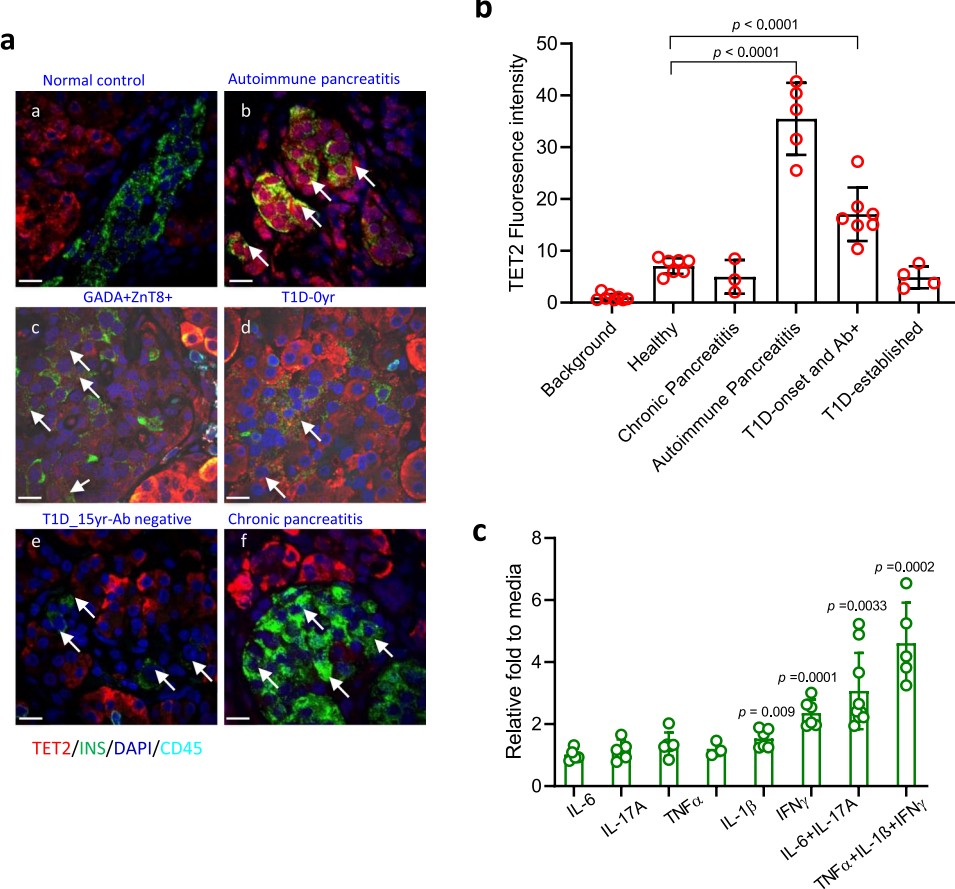

**Fig. 2 Induction of *TET2* in human β cells with inflammation. a** Expression of TET2 protein in human β cells in vivo in pancreas from a healthy control subject (a), patient with autoimmune pancreatitis (b), a nondiabetic individual with autoantibodies to GADA and ZnT8 who is a relative of a patient with Type 1 diabetes (c), a patient with recent-onset T1D (d), patient with Type 1 diabetes for 15 years without detectable autoantibodies (e), and (f) a patient with chronic pancreatitis. The sections were stained with anti-TET2 (red) and anti-insulin (green) as well as anti-CD45 (Cyan) antibody. DAPI (blue) stains the cell nucleus. The merged staining is shown. Arrows indicate TET2 + β cells in b, c, and d and β cells in e and f. Data were from normal individuals (n = 8), donors with autoimmune pancreatitis (n = 5), nondiabetic donors who were autoantibody+ (n = 7), and C-peptide + patients with T1D of relatively short duration (n = 7). Scale bars: 25 μm. **b** Image J quantification of the fluorescence intensity of *TET2* staining in islets studied in Fig. 1a. Twenty islets per condition were looked at, with three–seven donors for each condition. Different clinical conditions were compared to a healthy individual. Statistical analysis was performed using one-way ANOVA with Dunnett's multiple comparisons test. **c** Human islets were cultured with individual or combined cytokines as indicated for 24 h before transcription analysis of *TET2* gene by qRT-PCR. The fold of *TET2* mRNA induced by cytokines relative to islets cultured in media alone is shown. Data were mean ± SD from five experiments, each with 2000–8000 islet equivalents (IEQ) from nondiabetic individuals (Fold vs islets cultured in media alone. Statistical analysis was performed using one-way ANOVA with Dunnett's multiple comparisons test.

there were fewer surviving β cells in the WT recipients ($r = −0.82$, $p < 0.0001$) that was not seen in the Tet2-KO recipients ($p = 0.1447$). It was possible that autoreactive cell development was impaired in the Tet2-KO recipients. However, we found a similar frequency of IGRP-reactive CD8+ T cells in the WT and Tet2-KO BM recipients (Fig. 3c). In addition, we transferred splenocytes from KO bone marrow transfer (BMT) recipients that did not have diabetes, 11–14 weeks after the BMT, or from diabetic WT BMT recipients, into NOD/*scid* mice. These NOD/*scid* recipients developed diabetes at a similar rate (Fig. 3d), indicating that autoimmune T cells are present in nondiabetic Tet2-KO BMT recipients.

To further understand the nature of the resistance of β cells to killing, we transplanted islets from 5–6-week-old KO vs WT mice under the kidney capsules of 6-week-old NOD WT recipients. Two weeks after the transplant, and in addition, gave $1 \times 10^7$ diabetogenic splenocytes to synchronize the development of diabetes. Recipients were followed for the development of hyperglycemia which required the killing of both the endogenous and the transplanted islets. The time to diabetes onset in the

recipients of KO islets was delayed by 4.5 weeks (median survival = 19.5 weeks) compared to recipients of NOD WT islets (median survival = 15 weeks) ($p < 0.0001$). All of the 11 recipients of WT islets developed diabetes whereas, at 25 weeks, three of the 12 recipients of KO islets were diabetes-free (Fig. 3e). When the graft-bearing kidney was removed, there was rapid hyperglycemia, indicating that the transplanted islets were the source of insulin in the mice and that the endogenous WT β cells in the recipients had been destroyed. The Tet2-KO β cells could be killed by chemical means (i.e., streptozotocin) in vivo (Supplementary Fig. 3), suggesting that the resistance of the KO β cells to killing was to immune mechanisms.

To directly assess the effects of inflammatory mediators on Tet2-KO β cells, we cultured single islet cells with inflammatory cytokines that cause β cell death and found increased survival of Tet2-KO vs WT β cells (Fig. 3f). The resistance of the KO β cells was greater than α cells (Fig. 3f). Second, we cultured WT NOD/*scid* and Tet2-KO β cells with purified islet infiltrates from prediabetic WT NOD mice and found the greater killing of the WT vs Tet2-KO β cells (Fig. 3g).

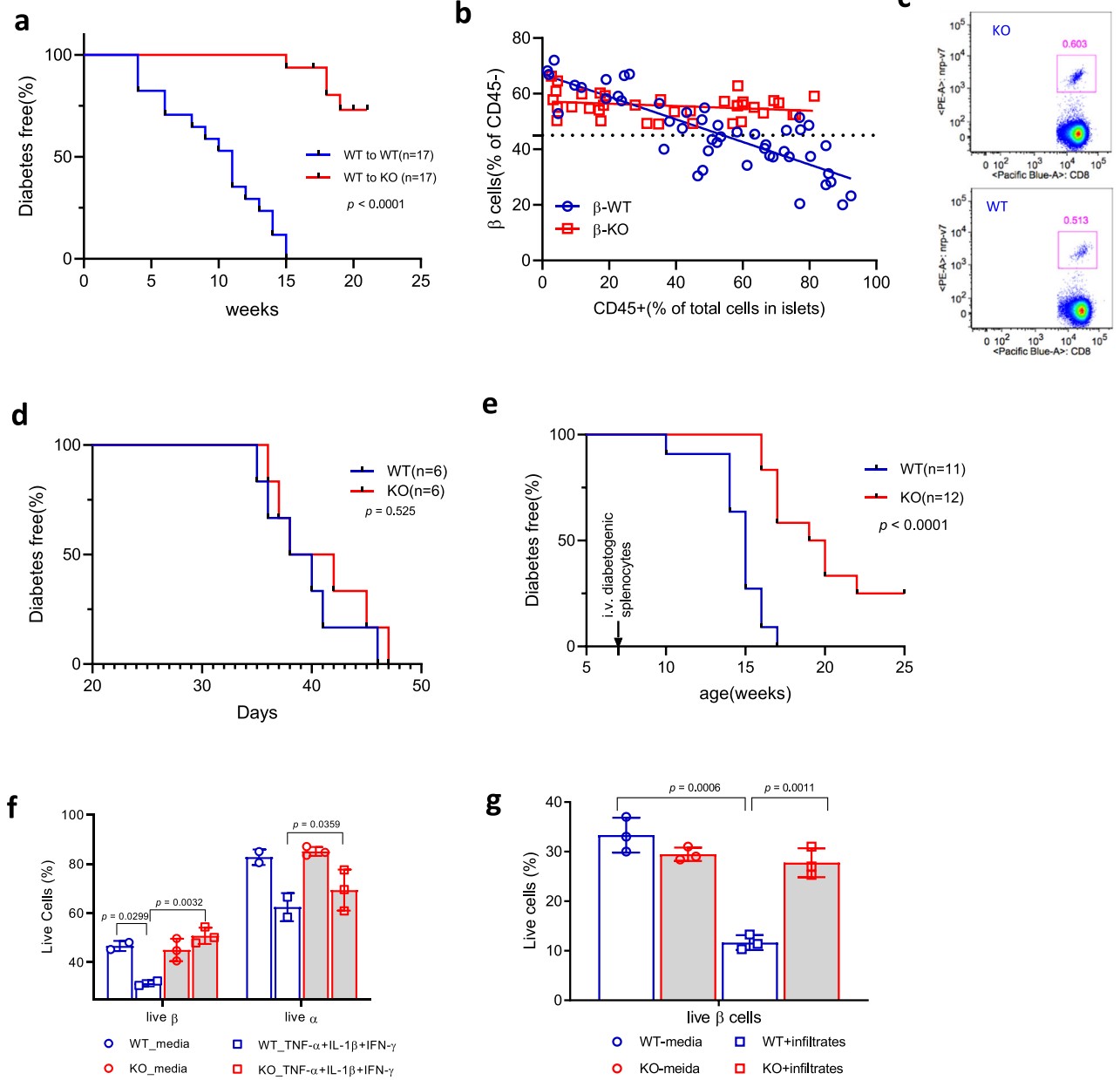

**Fig. 3 β cells lacking *Tet2* are protected from immune killing in vivo and in vitro. a** Diabetes incidence in WT or KO recipients of WT bone marrow. Statistical analysis was performed using Log-rank curve comparison. $n = 17$ for each group. **b** Relationship between cellular infiltrates and loss of β cells in WT and Tet2-KO recipients of WT bone marrow. The frequency of the intra-islet CD45 + cells and β cells (insulin+) analysed by FACS. Simple linear regression was used together with a two-side correlation test for statistical analysis. For WT recipients, Pearson $r = -0.82$, $p < 0.0001$; for Tet2-KO recipients, $p = $ ns. Each circle/square represents one mouse (WT, $n = 47$; KO, $n = 36$). **c** Frequency of IGRP-reactive CD8 + T cells in the spleen analysed by FACS. Fourteen weeks after transplant, splenocytes from KO or WT BMT recipients were stained with NRPV7 tetramer to identify IGRP-reactive CD8 + T cells. A single pair, representative of three experiments are shown. **d** Transfer of diabetes into NOD/*scid* mice with $10^7$ splenocytes from either nondiabetic KO or hyperglycemic WT BMT recipients 12 weeks post-BMT. Data were from two experiments, each with pooled splenocytes from two KO and two WT BMT recipients. Statistical analysis was performed using Log-rank curve comparison. **e** Diabetes incidence in islet transplant recipients from either WT or Tet2-KO NOD. About 250–300 pooled islets from 5–6-week-old WT or KO NOD mice were transplanted under the kidney capsule of 6-week-old WT NOD followed by transfer of $10^7$ diabetogenic splenocytes. The median survival time in WT is 15 vs 19.5 weeks in KO recipients. Statistical analysis was performed using Log-rank curve comparison (WT, $n = 11$; KO, $n = 12$). **f** Islet β cell survival during culture with cytokines. Single islet cells from WT or Tet2-KO B6 mice were cultured with the indicated cytokines for 12 h and the live cell percentage was determined in β as well as α cells (identified by insulin and glucagon staining) by FACS. Results are mean ± SD of three experiments, $n = 6$ each. Statistical analysis was performed using two-way ANOVA with Tukey's multiple comparisons test. **g** β cell survival during culture with diabetogenic islets infiltrates. Sorted β cells from either WT NOD/*scid* or Tet2-KO NOD mice were cultured overnight with islets infiltrates sorted from the same prediabetic WT NOD at a ratio of 1:5. Results are mean ± SD of three experiments, $n = 5$–6 each. Statistical analysis was performed using one-way ANOVA with Dunnett's multiple comparisons test.

These initial observations suggested that a subpopulation of β cells that survive immune attack have reduced *Tet2* despite the general increase in expression that we had observed during disease progression in vivo, and in human β cells that were exposed to inflammatory cytokines. In our previous studies in NOD mice, we identified a subpopulation of β cells, characterized by reduced insulin granularity, increased expression of "stem-like" genes, and reduced β cell markers such as *Pdx1*, *Nkx6.1*, and *MafA* that was resistant to immune-mediated killing[15]. Therefore, we analyzed Tet2 expression among these subpopulations of β cells from NOD mice. We found that *Tet2* levels were higher in the normal (i.e., "top") β cells that we showed succumbed to immune killing during the progression of the autoimmune disease when compared to the hypogranular (i.e., "bottom") cells that survived cytokine and immune cell-mediated killing (Supplementary Fig. 4).

**Immune responses are modulated by *Tet2*-deficient islet cells.** We analyzed the islet infiltrates from the WT and KO recipients of WT BM, 8 weeks after transplant to understand how the Tet2-KO islet cells changed the immune responses. The frequency of infiltrating cells was similar in the KO and WT islets (Supplementary Fig. 5). We analyzed gene expression by NanoString (pan-immunology panel) and found differential expression of 189 of 770 genes. There was decreased expression of genes associated with kinase signaling needed for T cell activation such as Stat3, Jak2, Smad3, Nfkb, and Mapk, reduced expression of molecules associated with cell–cell interaction, as well as T cell function such as Cd137, Gzmm, and Gzmk in addition to a panel of reduced cytokines (Fig. 4a). In addition, the infiltrating cells from KO recipients show increased expression of pro-cell death genes such as Bax, Casp3, Casp8, and Tnfsf10 (Trail) and reduced pro-survival gene Bcl2. Pathway analysis (with Ingenuity Pathway Analysis (IPA)) identified differences in inflammatory responses implicating 91 genes. Other affected pathways included cell-to-cell signaling and interaction, cellular function and maintenance, growth and proliferation, and movement which were significantly decreased in immune cells from KO recipients. Th1 and Th2 activation pathways were reduced in KO recipients as well as pathways of cell trafficking (Table 1). There were phenotypic difference between the infiltrating T cells in the KO and WT mice. To confirm these findings, we analyzed immune cells from the draining (pancreatic) lymph nodes from KO and WT BMT recipients of WT BM by FACS. The T cells from the KO recipients showed reduced frequencies of total T cells and increased frequency of B cells (Fig. 4b). In addition, there was reduced expression of CCR7+ and CXCR3+ CD4 T and CD8 T cells and fewer central memory CD4+ T cells (Fig. 4c), suggesting that pathologic (i.e., CXCR3+) T cells are not recruited in the KO recipients. On diabetes antigen-(IGRP) specific T cells in the pLNs (Fig. 3c), we likewise found reduced expression of CXCR3. Other activation markers like CD44 and PD1 were similar between KO vs WT recipients (Fig. 4d).

**β cells lacking Tet2 show reduced inflammatory responses.** The differences in WT and KO β cells might affect the infiltrating immune cells. Therefore we compared β cells (Zn + TMRE+) from the WT or KO BMT recipients 8 weeks after WT transplant and analyzed the cells by RNA-seq. About 333 genes were found to be differentially expressed between KO and WT β cells ($p < 0.05$ after FDR correction (Supplementary Fig. 6)), and 199 of these genes showed ≥1.5-fold change in expression (Fig. 5a, b). We identified a number of differences in pathways associated with β cell death and immune protection. These included reduced PTEN signaling ($p = 0.00129$)[25,26], inflammatory mediators (*Irf8*

and *Tifa*), attenuated immune signaling (*Pik3cd* and *Abl2*), and cytokine-inducible genes (*Gbp2*, *Gbp6*, *Ifi47*, and *Tnfaip2*). Genes involved in Class I and II MHC expression were reduced but the levels of Class I MHC were not reduced by FACS on β cells from Tet2-KO mice (Supplementary Fig. 7). Meanwhile, β cells from KO recipients expressed genes associated with immune suppression (GABA receptor A (*Gabrg3*)[27], *Gad2*, *Cd14*[28], and *H2-Q7* (Qa-2)[14,15] (Fig. 5a, b). Compare to WT β cells, there were changes in the expression of genes related to improved β cell fitness (*Slc39a4*, *Tgfbr2*, *Fgfr2*, *Fgfr3*, and *Aldh1a3*)[29–32] and survival (*Gas6*, *Ptpn13*, and *Tnfrsf19*)[33–37] (Fig. 5a, b).

To understand the action of *Tet2* in β cells, we performed ATAC-seq on the same β cell samples analyzed by RNA-seq in Fig. 5a, b. We identified 60,449 open chromatin sites among the five samples profiled. Accessibility at the overwhelming majority of these sites (~97%) did not differ substantially between WT and Tet2-KO cells, suggesting that *Tet2* deficiency does not lead to widespread chromatin remodeling of β cells. About 715 sites (~1%) had greater accessibility in KO's (FC >1.5) and 1387 (~2%) had lower accessibility (FC < −1.5) in Tet2-KO vs WT cells (Fig. 5c). Using HOMER transcription factor (TF) motif enrichment analysis, 97 TF motifs were enriched in peaks with lower accessibility (closing peaks) in KO β cells (*q* value <0.05) (Supplementary Table 2). Interestingly, motifs for 15 TFs mediating inflammatory responses such as Stats 1, 3, 4, and 5 and Irfs 1, 2, 4, and 8 (Fig. 5d) were specifically enriched in closing peaks, suggesting that inflammatory stress-responsive *cis*-regulatory elements are epigenetically decommissioned in Tet2-deficient β cells. Fifteen TF motifs were identified as enriched in peaks with higher accessibility (opening peaks) in KO β cells (*q* value <0.05). Most of these corresponded to TFs controlling islet β cell identity and function such as Foxa1, Foxa2, and Rfx and were also enriched in closing peaks in KO β cells (Supplementary Table 3), indicating that *Tet2* deficiency results in opening and closing of a subset of islet TF binding sites.

To confirm these findings and the responses of β cells to inflammatory mediators, we first screened inducible cytokines and chemokines in islets following IFNγ with or without TNFα + IL-1β. *Il6*, *Cxcl10*, *Cxcl16*, *Ccl2*, as well as *Tnf* were induced by this cytokine cocktail (TNFα + IL-1β + IFNγ) but not *Ccl12*, *Ccl19*, or *Il1b* (Supplementary Fig. 8a).

Previously Xu et al. described impaired STAT1 signaling in tumor cells cultured with IFNγ[18]. Similarly, we identified reduced IFNγ response measured by the production of *Cxcl10*, *Cxcl11* as well as the expression of *Pdl1 but not Cxcl9* in KO vs WT islets from B6 mice (Supplementary Fig. 8b). We cultured WT and KO islet cells with TNFα + IL-1β+IFNγ and found in Tet2-KO islets reduced induction of *Il6*, *Ccl2*, *Cxcl10*, *Cxcl16*, and *Tnf*, which have been linked to β cell destructive processes (Fig. 6a). Islets from WT or KO B6 mice were used to avoid any immune cell effects. In addition, we found that the expression of genes associated with β cell death such as *Fas*, *Tnfsf10b* (TRAIL-R2), and *Stat1*[38,39] were also reduced in cultured KO β cells (Fig. 6b). DNA from β cells were recovered from KO and WT BMT recipients and the enrichment of 5-hydroxymethylcytosine (5hmC) on the promoter regions (500 bp within TSS that contains multiple CpG sites) of genes measured in Fig. 6a, b was compared. Promoter regions of *Il-6*, *Cxcl10*, *Cxcl16*, *Fas*, and *Stat1* show reduced 5hmC levels in KO vs WT β cells (Fig. 6c). When human islets were cultured with cytokines, we found a similar reduction of chemokine, cytokines as well as death receptors as seen in the mouse experiment when *TET2* inhibitor R-2HG[40,41] was added to the culture (Fig. 6d).

Finally, to determine whether these differences in inflammatory responses are associated with β cell survival, we transferred splenocytes from diabetic NOD mice into irradiated WT and KO

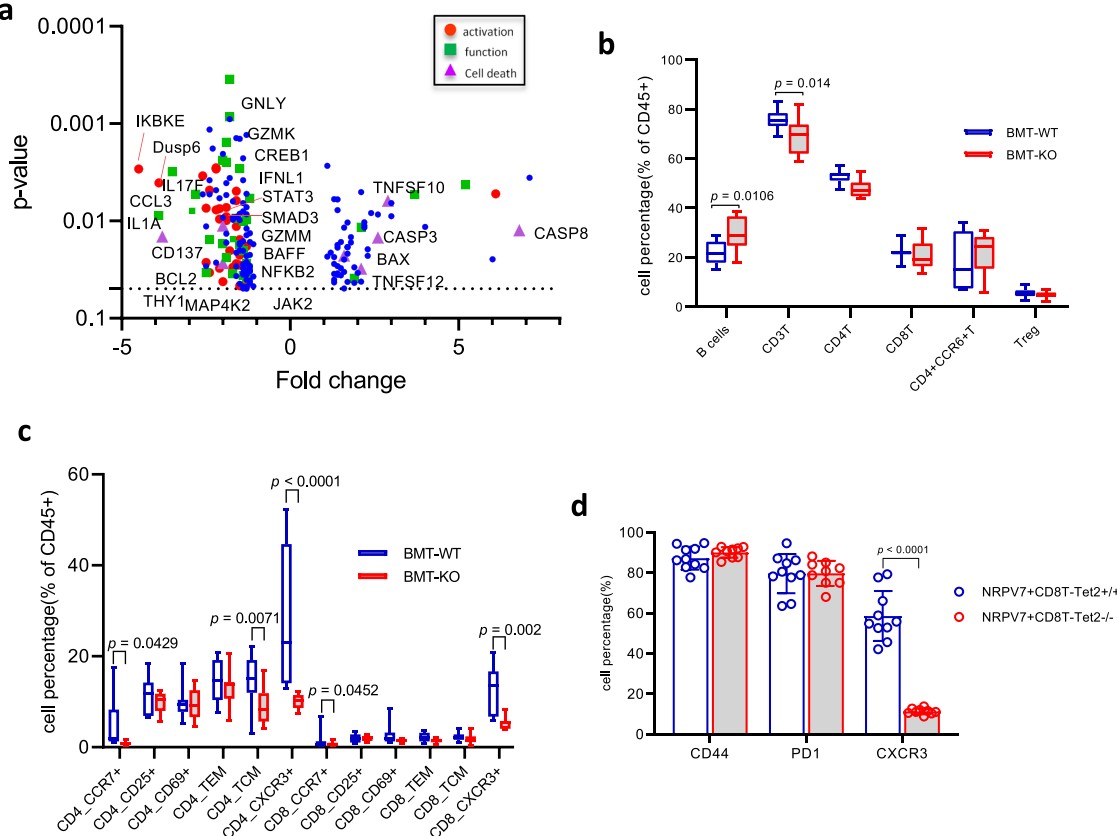

**Fig. 4 T cells from Tet2-KO islets are less activated and show reduced pathogenic phenotypes. a** Volcano plot showing transcription differences in islets infiltrates between KO and WT recipients of WT bone marrow. Eight weeks post-BMT, CD45+ islets infiltrates were sorted from KO vs WT recipients and used subsequently for transcription analysis with NanoString Pan-Immunology panel. The data were analyzed with nSolver4.0 analysis software. 189 of 770 genes that were found to be different between infiltrates from WT and KO recipients are shown (after FDR correction, $p < 0.05$). Genes that are associated with T cell activation, cell function as well as cell death are highlighted with different colors among the 189 genes. Data were from six KO vs six WT samples that were pooled from nine KO and eight WT mice, respectively. **b**, **c** FACS analysis of CD45+ cells from pancreatic lymph nodes (pLNs) from KO or WT BMT recipients. The % of CD45+ cells is shown. (Treg = CD4 + CD25 + CD127$^{lo}$, TCM = CD44 + CD62L$^{hi}$, TEM = CD44 + CD62L$^{lo}$). Data were from four–five experiments with pLNs from individual mice. The box indicates 25–75%. The whiskers show the minima to the maxima values and the central line indicates the median. Statistical analysis was performed using two-way ANOVA with Sidak's multiple comparisons test (WT: $n = 11$ in (**b**), 12 in (**c**); KO: $n = 14$ in **b**, 11 in **c**). **d** FACS analysis of the activation status of NRPV7 tetramer-positive IGRP-reactive CD8+ T cells from pLNs of KO and WT BMT recipients. Data were from three independent experiments and each data point represents an individual mouse. Statistical analysis was performed using two-way ANOVA with Sidak's multiple comparisons test.

**Table 1 Diseases and bio functions identified by Ingenuity Pathway Analysis of Nanostring data.**

| Diseases and disorders | $p$ value range | #Molecules |
|---|---|---|
| Inflammatory response | 1.22E-04 – 9.69E-49 | 91 |
| Immunological disease | 8.60E-05 – 2.59E-26 | 44 |
| **Molecular and cellular functions** | $p$ value range | #Molecules |
| Cellular-to-cell signaling and interaction | 1.22E-04 – 9.69E-49 | 86 |
| Cellular function and maintenance | 1.22E-04 – 1.43E-45 | 85 |
| Cellular development | 1.22E-04 – 1.58E-42 | 82 |
| Cellular growth and proliferation | 1.22E-04 – 1.58E-42 | 84 |
| Cellular movement | 1.22E-04 – 8.71E-41 | 62 |
| **Physiological system development and function** | $p$ value range | #Molecules |
| Immune cell trafficking | 1.22E-04 - 9.69E-49 | 90 |
| Lymphoid tissue structure and development | 1.22E-04 - 1.58E-42 | 92 |
| **Top canonical pathways** | $p$ value | Overlap |
| Th1 and Th2 activation pathway | 9.14E-21 | 15.2% 23/151 |
| Th1 pathway | 8.92E-19 | 18.1% 19/105 |

Diseases and bio functions identified by Ingenuity Pathway Analysis of PanCancer Immune Profiling Data. CD45 + infiltrates were sorted from KO vs WT recipients 8 weeks post-BMT from WT NOD BM donors ($n = 13$–16 KO or WT mice over six separate sortings). About 9000–45,000 cells were lysed for each reaction. Results were summarized into different disease and bio functions (marked in bold). The name of the specific pathway was listed as well as the $P$ value and number of molecules involved in each pathway.

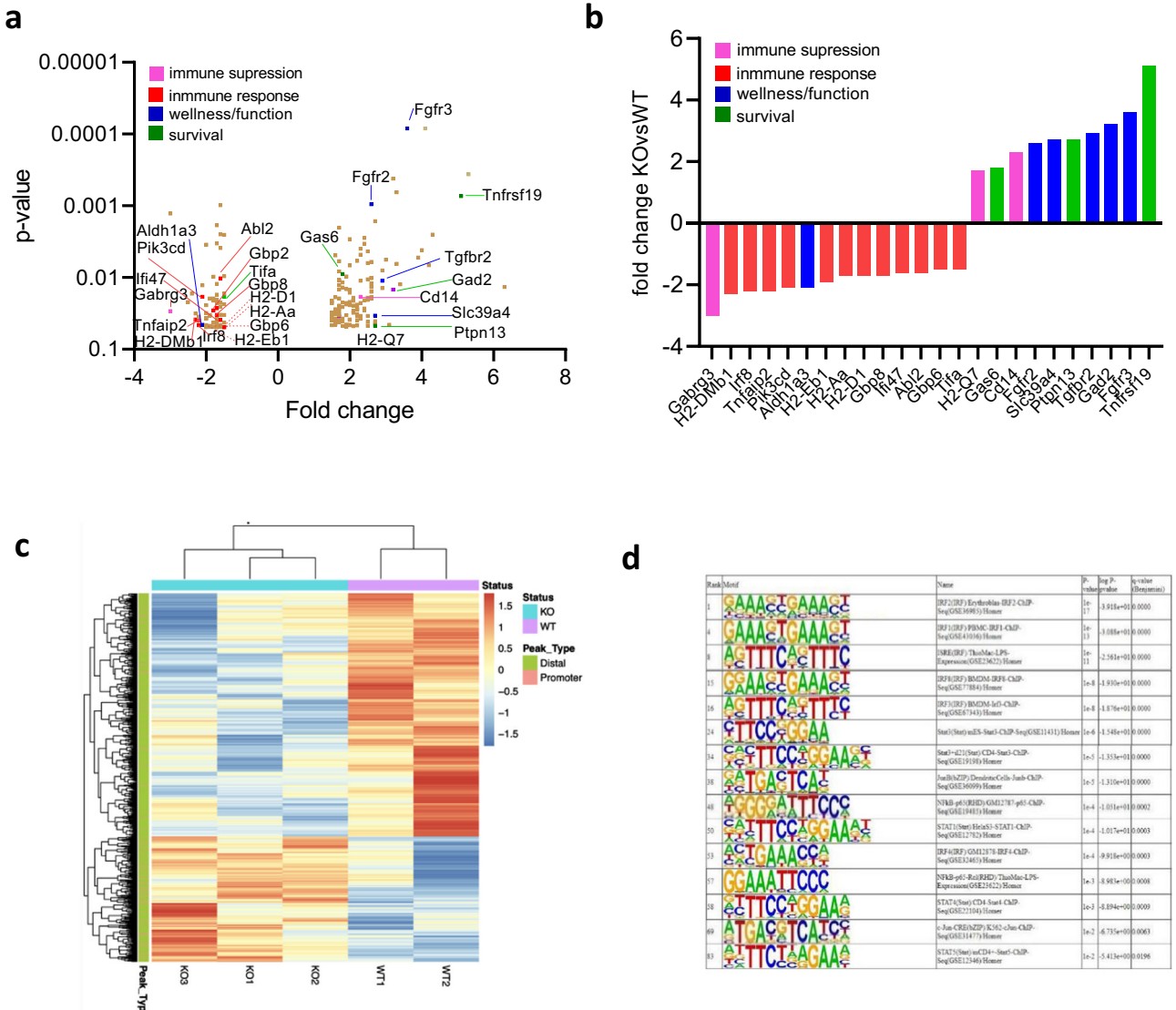

**Fig. 5 β cells lacking Tet2 have reduced inflammatory responses. a** Volcano plot showing gene expression differences in β cells from KO vs WT recipients of WT bone marrow. Eight weeks after the bone marrow transfer, β cells that are TMRE + Zinc+ were sorted from KO vs WT recipients and subjected to the cDNA library preparation and bulk RNA-seq analysis. There are 333 differentially expressed genes in total with p < 0.05 after FDR correction, and 199 with fold change >1.5 that are shown in **a**. Genes of particular interest are color-coded in **a** and **b** according to their function. **c** Comparison of ATAC-seq profiles identify chromatin accessibility changes in β cells from KO vs WT recipients of WT bone marrow. Of 60,449 accessible sites identified across the five samples, 715 (~1%) had increased accessibility (FC >1.5) and 1387 (~2%) had reduced accessibility in (FC < −1.5) in KO's, respectively. **d** HOMER motif analysis identified 97 TF motifs enriched in peaks with decreased accessibility (closing peaks) in KO β cell samples (q value <0.05), 15 of which are inflammatory mediators. The rank of the 97 TF motifs was sorted by q value. Data were from three sortings and three–four mice per group per sorting.

recipients and followed them for the development of diabetes. The median time to hyperglycemia was 9.5 weeks in the WT recipients whereas none of the six KO recipients developed diabetes for more than 20 weeks after adoptive transfer (p = 0.0041) (Fig. 6e).

## Discussion

We have shown that TET2 can control immune-mediated destruction of human and murine β cells through the interactions with immune cells and β cell-intrinsic mechanisms. *Tet2* expression is increased during the progression of diabetes in islet cells and in β cells. However, a subpopulation of β cells that we previously showed was protected from autoimmune killing, has lower levels of *Tet2* expression. In human pancreases, β cells also express *TET2* under inflammatory conditions, but islets from

normal individuals and from patients with long-standing T1D have β cells without detectable *TET2* expression. Our findings indicate that *TET2* is associated with, but not sufficient for, killing of β cells, since we also found increased *TET2* expression in β cells from patients with autoimmune pancreatitis who do not develop diabetes. Instead, our studies in Tet2-KO mice indicate that in the absence of Tet2, β cells are protected from immune-mediated killing. We used three complementary model systems: BM transplantation, islet transplantation, and adoptive transfer of diabetogenic splenocytes to identify the interactive relationships between Tet2 expression in β cells and immune cells and show that Tet2-KO islets are resistant to killing even in the presence or transfer of diabetogenic T cells. The Tet2-KO β cells show transcriptional differences, compared to WT in responses to inflammatory cytokines in genes associated with β cell wellness,

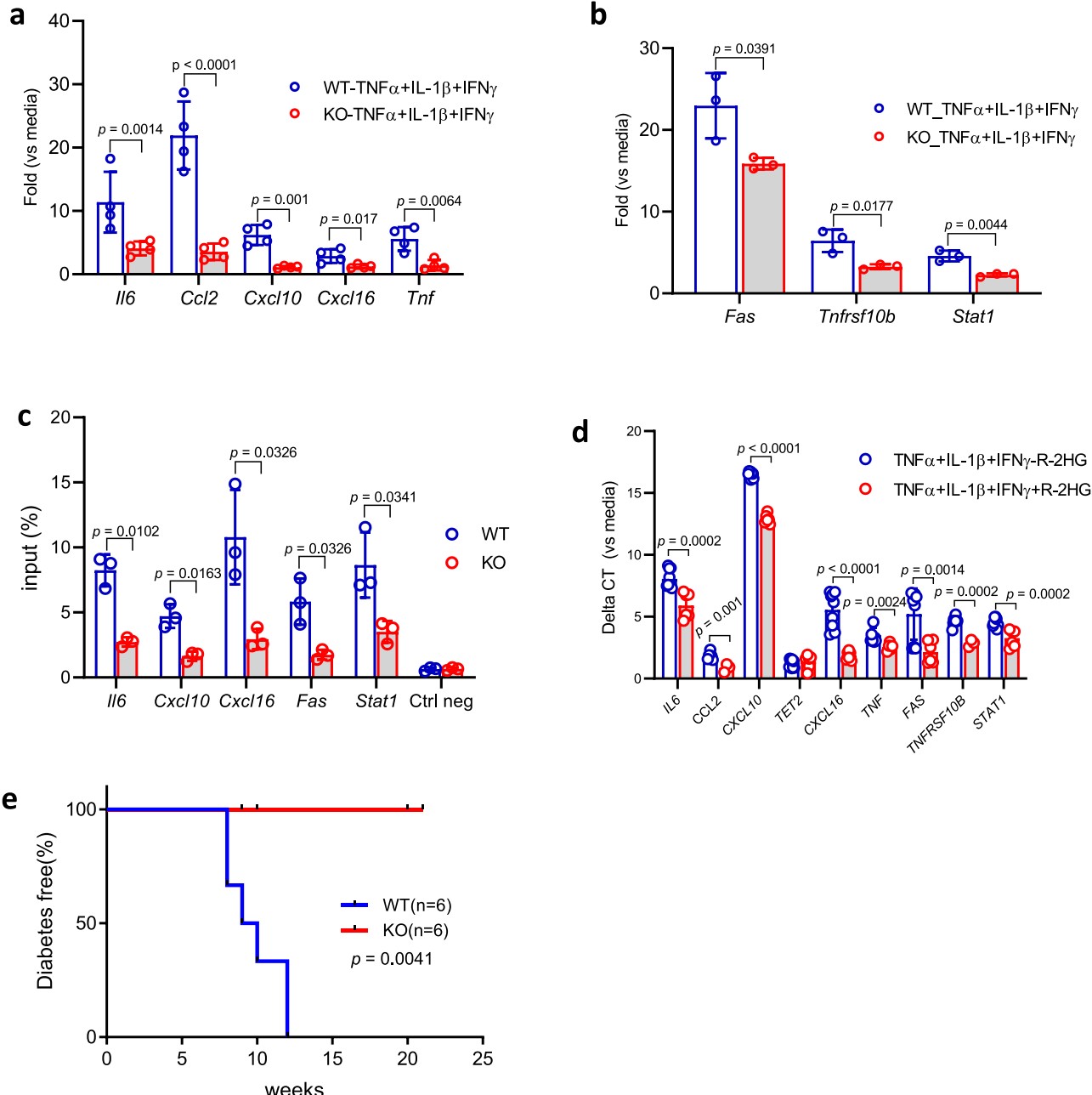

**Fig. 6 Tet2-KO ß cells show reduced inflammation response in vitro and are resistant to killing by diabetogenic cells in vivo.** Transcription analysis of cytokines and chemokines (**a**) as well as cell death receptors (**b**) in KO vs WT islets from B6 mice following culture with the indicated cytokines for 24 h. Data were mean ± SD of three–four independent experiments, four mice per group per experiment. Statistical analysis was performed using a two-tailed unpaired *t*-test without correction for multiple comparisons. **c** Enrichment of 5hmC at the promoter regions of indicated genes was examined by hMeDIP-qPCR. Comparisons were made between β cells from KO vs WT NOD recipients of WT NOD bone marrow. Data were mean ± SD of three independent experiments, three–five mice per group per experiment. Statistical analysis was performed using a two-tailed unpaired *t*-test without correction for multiple comparisons. **d** Transcription analysis of the genes studied in (**a** and **b**) in human islets after culture with TNFα +IL-1β + IFNγ with or without TET inhibitor R-2HG for 24 h. The Ct of cytokine group with or without R-2HG was compared to islets in media supplemented with or without R-2HG, respectively. Data (mean ± SD) were from three experiments, each with 4000–8000 islet equivalency (IEQ) from each nondiabetic donor. Statistical analysis was performed using a two-tailed unpaired *t*-test without correction for multiple comparisons. **e** Diabetes incidence in Tet2-WT or KO NOD mice that received by adoptive transfer 1.5 × 10⁷ diabetogenic splenocytes from diabetic WT NOD mice after half lethal dose irradiation (650 rads). Mice were followed up to 21 weeks post adoptive transfer. Statistical analysis was performed using Log-rank curve comparison. *n* = 6 for each group.

function, survival, and immune suppression, and reduced expression of genes associated with immune responses and activation. Importantly, there was also reduced chromatin accessibility at putative regulatory elements containing binding motifs of inflammatory mediators in Tet2-KO β cells. These molecular changes were confirmed by functional studies that showed

reduced secretion of cytokines and chemokines when the Tet2-KO islets were cultured with inflammatory cytokines. As a result, Tet2-KO islets fail to activate immune responses and autoreactive T cells, resulting in reduced autoimmune elimination of Tet2-KO ß cells compared to WT. In summary, our studies have identified a mechanism that controls interactions between β cells and

immune cells that leads to immune-mediated killing and suggest that inhibiting this pathway may lead to improved survival of β cells in the setting of autoimmunity.

In human pancreatic sections, we found differences in *TET2* expression within islet cells and between pancreases from individuals with different stages of β cell destruction. We found expression in both the nuclear and cytoplasm, consistent with reports in neurons[42]. In patients with long-standing T1D, the absence of TET2 expression suggests an association with β cell survival or possibly β cells that do not respond to inflammatory mediators.

Our studies of the control of β cell responses by Tet2 provide insight into the mechanisms that account for these histologic findings. A fully pathologic autoimmune repertoire developed in the Tet2-KO recipients of WT BM. However, these cells did not receive the same activation signals and did not cause β cell killing as shown by our analysis of their transcriptome, when compared to islet infiltrates from the WT BMT recipients. The absence of *TET2* expression on β cells, therefore, could potentially explain the ability of some β cells to survive even when autoreactive T cells remain in the host. The Tet2-KO BM recipients developed diabetogenic T cells since splenocytes were able to transfer diabetes to NOD/*scid* recipients at a rate similar to WT cell donors. Consistent with these findings, islets from the Tet2-KO mice showed significantly delayed rejection compare to islets from WT donors when they were transplanted into WT NOD recipients. Even in the same mouse, the WT but not KO islets were preferentially destroyed.

*Tet2* deficiency may affect the survival of β cells through direct or indirect pathways. ATAC-seq revealed reduced chromatin accessibility in *Tet2*-deficient ß cells at putative binding sites for Stat and Irf TFs that have been associated with cytokine signaling. Gene expression may occur from local epigenetic modifications and therefore decreased expression is expected with Tet2 deficiency[43]. Previous studies of Tet2-deficient tumor cells identified impaired STAT1 signaling that prevented the expression of chemokines and PD-L1 expression and enabled tumors to evade anti-PD-L1 immune therapy[18]. These investigators described a disrupted IFNγ-JAK-STAT signaling cascade in Tet2-deficient tumor cells that we also found with β cells.

These observations indicate that the islet cells themselves deliver signals to the autoreactive immune cells that leads to their activation and β cell killing. These signals most likely are the result of differences in the Tet2-KO β cells since the antigen-presenting cells in the transplant recipients were from the WT BM donor. Our data, in Supplementary Fig 8, show that IFNγ inducible chemokines (CXCL10 and CXCL9), as well as PD-L1, are reduced in the KO β cells, consistent with the inaccessibility of STAT1 signaling (Fig. 5d). Groom et al. described a CXCR3 inflammatory loop in which the IFNγ-induced chemokines recruit CXCR3-expressing T cells to inflammatory sites[44]. The production of IFNγ by the CXCR3+ T cells can provide a "feed forward" in which additional CXCR3-expressing inflammatory cells, such as those thought to mediate autoimmune diabetes are recruited. Thus, the inability to respond to IFNγ may block this loop and result in the reduced frequency of the pathological T cells[23,45–48]. A unique feature of this mechanism of protection is that it does not dependent on the specificity or the ability of the immune cells to respond to the target β cells per se but rather on the ability of the β cells to respond to inflammatory mediators and recruit and activate the T cells. In addition, as a result of the loss of Tet2, islet cells express reduced levels of molecules that are directly responsible for killing including Fas and Tnfrsf10b.

In addition to the participation of the β cells in immune cell interactions, we also identified intrinsic pathways to account for enhanced β cell survival in Tet2-KO β cells. We found improved survival when β cells were cultured with inflammatory cytokines as well as infiltrating immune cells, both of which may mediate β cell killing. In the Tet2-KO β cells, expression of GABA receptor A (*Gabrg3*), which is associated with immune suppression, was reduced and there was increased expression of *Tnfrsf19*, *Fgfr2*, and *Fgfr3*. Increased *Tnfrsf19* expression was shown to attenuate TNFβR signaling and improve β cell survival[29,30]. Fgfr2 and Fgfr3 are growth factor receptors, and attenuation of FGF signaling by expressing dominant-negative forms of the FGFRs receptors in mouse β cells leads to diabetes[31]. Gas6, which is linked to β cell proliferation, was also increased in the expression on the KO β cells[49]. These cell-intrinsic mechanisms may account for the resistance of the cells to killing by inflammatory cytokines in the absence of extrinsic cells.

There are a number of limitations of our studies. First, *TET2* loss of function mutations occurs frequently in humans[18]. Therefore, the applicability of our findings in the inbred NOD mouse needs further confirmation. Moreover, some have questioned the model's relevance to human disease since there are a number of nonspecific interventions that may modulate spontaneous diabetes in this model[50]. *Tet2* causes demethylation of methylcytosines, and therefore the relationship between the loss of this enzymatic activity and decreased transcription of genes is somewhat paradoxical. However, our previous studies showing increased activity of Dnmts suggest a mechanism that, if unopposed, could lead to these outcomes[51]. In our *Tet2*-deficient NOD mice and recipients of bone marrow transplants (BMTs), *Tet2* was absent throughout the development of immune cells. We do not know if there is a critical time(s) when *Tet2* expression is involved in β cell killing as well as immune activation but our adoptive transfer studies with diabetogenic T cells would suggest that absence of *Tet2* can be protective even when β cells are attacked by fully matured effector cells. In addition, the relationship, particularly in human tissues, between the autoreactive repertoire, *TET2* expression, and β cell killing will require further studies with human β and effector T cells. Lastly, there may be an effect of other environmental factors, on the activation of T cells in the *Tet2*-deficient mice, such as the microbiome that may affect their activation[52].

In summary, our studies have identified an interaction between β cells and immune cells that regulates immune cell activation, recruitment, and killing. Expression of Tet2 is required for β cells to produce chemokines and cytokines that are needed for the activation and pathologic function of diabetogenic T cells. In its absence, β cells resist cell- or cytokine-mediated killing even when potentially autoreactive cells are present through cell-intrinsic mechanisms as well as avoiding activation of immune cells. These studies suggest a potential future strategy to protect insulin-producing cells from immune-mediated killing in the setting of autoimmunity or even for engineering more resilient and/or resistant β cell replacements to avoid recurrent immune elimination.

## Methods

**Mice**. C57BL/6[Tet2−/−] male breeder under B6 background was purchased from The Jackson lab and backcrossed with NOD females for over 14 generations. Female NOD, B6, and NOD/*scid* mice were also obtained from The Jackson Laboratory and maintained under pathogen-free conditions. All protocols were approved by the Yale Institutional Animal Care and Use Committee. Animals were housed under the following facility conditions: Temperatures: 21–23 °C; Humidity: 50–60%. Lighting: 12:12 (i.e., 12 h of light from 7 a.m.–7 p.m. then 12 h of dark from 7 p.m.–7 a.m.).

**Intraperitoneal glucose tolerance test (IPGTT)**. B6 Tet2-WT, HET, and KO mice fasted for 16 h. Blood glucose levels were measured 0, 15, 30, 60 and 120 min after glucose injection (2 g/kg body weight). IPGTT result was presented as the area under the curve using trapezoidal rule[53] and then divide by the total assay time (120 min).

**Human islets and pancreas tissues**. Human islet samples were obtained from adult, nondiabetic organ donors from the Prodo Labs (details listed in Supplementary Table 1). Pancreas tissues from donors with autoimmune pancreatitis ($n = 5$) or chronic pancreatitis ($n = 3$) were obtained from the Department of Pathology at Yale and histologic slides of the pancreas were obtained from normal individuals ($n = 8$), nondiabetic donors who were autoantibody+ ($n = 7$), and C-peptide+ patients with T1D of relatively short duration ($n = 7$) from nPOD. The use of human tissues and islets was considered "not human subjects research" as per institutional policies (Yale Institutional Review Board). It did not involve data obtained through intervention or interaction with the individual and did not contain identifiable private information.

**Immunofluorescent labeling of human pancreas**. To identify TET2 protein expression in pancreatic islet cells, formalin-fixed, paraffin-embedded sections (5-µm thickness) of human pancreas were incubated with primary antibodies against TET2 (Abcam, cat# 230358), insulin (DAKO, cat# A0564), and CD45 (DAKO, clone# 2B11 + PD7/26) overnight at 4 °C, washed and processed with the appropriate secondary fluorescent-conjugated antibodies. Subsequently, sections were dyed with 0.7% Sudan Black and CuSO4 to quench auto-fluorescence and counterstained with DAPI[54]. Images were acquired on an UltraVIEW VoX (Perkin-nElmer) spinning disc confocal Nikon Ti-E Eclipse microscope using the Volocity 6.3 software (Improvision). Image J was used to quantify the fluorescence intensity of TET2 staining in the islets in different clinical conditions. Twenty islets were picked per condition was from three–seven donors.

**Mouse islet isolation and β cell staining and purification**. Mouse islets were handpicked with a stereomicroscope after collagenase digestion of pancreases and single-cell suspensions were prepared[15,17]. For β cell enrichment, single islet cells were stained with TMRE and FluoZin-3 and sorted with a FACSAria II (BD). In some experiments, intracellular staining with anti-insulin (R&D Systems, Clone # 182410), anti-Tet2 (Cell Signaling Technologies, clone# D6C7K), and anti-glucagon (Abcam, clone# K79bB10) was used at 2.5 mg/ml to identify β cells or α cells and analyzed or sorted with an LSRFortessa (BD).

**Flow cytometry on immune cells**. For standard surface staining, lymphocytes ($10^6$ cells/sample) were washed with PBS and incubated for 30 min at 4 °C (dark) in 100 µl PBS plus 2% FBS with indicated fluorochrome-labeled monoclonal antibodies. After washing with PBS two times, 30,000–50,000 live cells were analyzed by flow cytometry. For surface staining, CD45 (clone# 30-F11, cat#103125), CD4 (clone#GK1.5, cat#100411), CD8 (clone#53-6.7, cat#100727), CD25 (clone#3C7, cat#101903), CD69 (clone#H1.2F3, cat#104505), CXCR3 (clone#CXCR3-173, cat#126535), CCR7 (clone#4B12, cat#120113), CD44 (clone#IM7,cat#103015), and CD62L (clone#MEL-14, cat#104407) were diluted to a working concentration of 2 mg/ml. All antibodies were from BioLegend. Gating strategies are included as Supplementary Fig. 9.

**Islet cultures**. Human or mouse islets were rested upon arrival or isolation in complete RPMI media for 3 h before aliquoting (100 per well) into a non-treated 24-well plate and culture for 18–24 h with or without cytokines. All mouse and human cytokines were used at 10 ng/ml, except for human IL-6 at 20 ng/ml and human IL-17A at 50 ng/ml. TET inhibitor R-2HG was used at 400 nM. The cells were then harvested for RNA extraction and transcriptional analysis of genes of interest. In cytokine or immune cell-mediated killing experiments, mouse islets were dispersed with accutase and cultured as single cells overnight before harvest for live/dead analysis by flow or cell death receptors analysis by qRT-PCR.

**Adoptive transfer of diabetogenic splenocytes**. Tet2-KO or WT NOD mice (8 weeks of age) were irradiated with a split dose ($2 \times 325$ rads, 2–3 h apart, 650 rads total) and splenocytes ($1 \times 10^7$) from a WT NOD mouse with recent-onset DM were transferred i.p. within 3 h of irradiation. The time to diagnosis of diabetes (i.e., glucose >250 mg/dl × 2) was recorded after the adoptive transfer.

**Streptozotocin (STZ) treatment**. Eight to 10-week-old Tet2-WT and KO B6 mice were given a single dose of STZ (200 mg/kg, i.p) and followed daily following the treatment.

**Bone marrow transplant (BMT)**. For BMTs, donor WT NOD mice 6–8 weeks of age were euthanized by CO2. BM was flushed with ice-cold HBSS and the erythrocytes were removed by lysis. The recipient KO or WT mice (4–8 weeks of age) were irradiated with split dose ($2 \times 550$ rads, 2–3 h apart, 1100 rads total) and donor BM ($10 \times 10^6$ cells) transferred i.v. within 3 h of irradiation[55]. Recipients were followed for hyperglycemia (glucose >250 mg/dl × 2) and diabetic incidence was recorded at the time after BMT.

**Islet transplant**. About 250–300 handpicked islets were pooled from 5–6 weeks KO or WT donors (two–three donors per recipient). They were transplanted under the kidney capsule of WT NOD recipients (6 weeks of age). Two weeks later,

diabetogenic splenocytes were given i.p.to the transplant recipients ($1 \times 10^7$ cells per recipient). Blood glucose was measured twice weekly and diabetes incidence was followed and recorded at the age of islets recipients.

**Low-input RNA-seq and data analysis**. Viable β cells were enriched by FACS sorting from KO or WT recipients 8 weeks post-BMT from WT BM donors. RNA was extracted and reverse transcription and cDNA amplification were performed using 10 ng of RNA using the SMARTer Ultra Low RNA kit (Clontech Laboratories) as previously reported[15]. Sequencing libraries were prepared using the Nextera XT DNA Sample Preparation kit (Illumina). Libraries from eight samples were pooled and sequenced with a $2 \times 100$ bp paired-end protocol on the HiSeq 4000 Sequencing System (Illumina), occupying half a lane with ~300 million reads. The sequencing was done at the Yale Stem Cell Center Genomics Core. The reads were mapped to mouse reference transcriptome mm9 and mRNA was quantified. RNA-Seq data were analyzed with Partek Flow software (v6.6).

**NanoString Pan-Immunology Panel Analysis of islet infiltrates**. Analysis was done with the NanoString Technologies Pan-Immunology panel as per the manufacturer's instructions. Briefly, 9000–45,000 CD45+ islet infiltrates were sorted from KO or WT BMT recipients and cell pellets were lysed in 33% RLT lysis buffer (QIAGEN). Up to 3.5 µl lysate was used directly in a 15 µl hybridization reaction which was kept at 65 °C for 24 h. The hybridization product was prepped and used with the digital analyzer nCounter MAX/FLEX system. The data were analyzed with nSolver4.0 analysis software (NanoString Technologies). Differentially expressed genes ($P$ value <0.05 after FDR correction) were identified and pathways were analyzed with IPA and upstream regulator analysis (https://www.qiagenbioinformatics.com/products/ingenuity-pathway-analysis/).

**RNA extraction and RT-PCR analysis**. RNA was extracted with RNeasy Plus Mini Kit (QIAGEN, cat#74134), and the High-Capacity cDNA Reverse Transcription Kit was used for cDNA synthesis (Applied Biosystems, cat#4368814). Primer pairs were used in this study together with a QuantiFast SYBR Green PCR kit (QIAGEN). The sequences of primer pairs used in this study are listed in Supplementary Table 4. The $Actb$ housekeeping gene was used to normalize the input RNA in all real-time qPCR assays. Gene expression is presented as Delta $Ct = (Ct$ of $Actb - Ct$ of target gene$) + 20$ or in other cases as relative fold to control. In Supplementary Fig. 1e, RNA was recovered from perm/fixed cells with RecoverAll™ Total Nucleic Acid Isolation Kit for FFPE (Invitrogen, AM1975). Taqman probes were used in qPCR[15].

**ATAC-seq and data analysis**. ATAC-seq libraries were prepared from FACS sorted mouse islet β cells[56]. After wash and lysis, the cell pellet was resuspended in 50 ul of transposition mixture (Illumina Tagment DNA TDE1 Enzyme and Buffer) and incubated for 30 min at 37 °C. Transposed DNA was cleaned with a Qiagen MinElute kit and stored frozen at −20 °C until indexing as decribed[57]. Libraries were sequenced on an Illumina NovaSeq S1 with $2 \times 101$ bp cycles. Raw sequence reads were quality trimmed using Trimmomatic version 0.32[58] with the following parameters "TRAILING:3 SLIDINGWINDOW:4:15 MINLEN:36." Trimmed reads were aligned to the mouse genome (mm10) using BWA version 0.7.12[59], specifically using the bwa mem –M option. Duplicate reads were marked and removed using "MarkDuplicates" from Picard-tools version 1.95 (http://broadinstitute.github.io/picard/). The quality of aligned reads was examined using Ataqv v.1.0.0[60]. After preprocessing and quality filtering, peaks were called on alignments with MACS version 2.1.0.20151222[61] using the parameters "-p 0.0001 -g mm -f BAMPE -nomodel -nolambda -B -keep-dup all -call-summits". Peaks located in blacklisted regions of the genome were removed. Overlapping peaks from all samples were merged with BEDTools version 2.26.0[62] resulting in 60,449 consensus peak regions. Raw read counts in these peaks for each sample were determined using the R package DiffBind_2.4.8[63]. To identify peaks with differing chromatin accessibility between WT and KO samples, the log mean normalized reads (log2 counts per million) were compared between the two sample groups. Peaks with a differing accessibility >1.5-fold in either group were used for TF motif enrichment analysis. The HOMER suite version 4.6[64] and "findMotifsGenome.pl" script with parameters "mm10 -size 200" was used to determine TF motifs enriched ($q$ value <0.05) in the peaks of interest.

**Hydroxymethylated DNA immunoprecipitation and qPCR**. The experiments were carried out using the hMeDIP kit (Active Motif, 55010) according to the manufacturer's instructions. Briefly, β cells were enriched from islets 8 weeks after BMT for genomic DNA recovery with the DNeasy Blood & Tissue Kit (QIAGEN, 69504), followed by digestion using restrictive enzyme Mse I (NEB, R0525S). Digested DNA was purified through with DNA purification column (Qiagen, 28104). Purified DNA (1 µg) was immunoprecipitated with 2.5 µg of mouse anti-5hmC antibody from the kit and was incubated overnight at 4 °C. The magnetic beads were added to the DNA–antibody mixture for 2 h incubation at 4 °C and isolation of immunoprecipitated DNA was performed according to the kit instructions. SYBR® Green Supermix (Bio-Rad) on a Bio-Rad CFX96 Real-Time system was used to perform qPCR, as indicated by the manufacturer's protocol. Eluted DNA was undergone qPCR as template and primers adjacent (500 bp to the

transcription starting site (TSS)) to the promoter regions of target genes was used for amplification to analyze the changes in the local 5hmC signal in the respective promoter regions. The Ct values of each sample were used in the post-PCR data analysis. The relative enrichments (after normalization against control IgG) of the indicated DNA regions were calculated using the Percent Input (10% of the total DNA used in the reaction) Method according to the manufacturer's instructions.

**Statistical analysis**. The number of samples per group and experiment repeats, as well as the statistical tests used, is indicated in each figure legend.

**Reporting Summary**. Further information on research design is available in the Nature Research Reporting Summary linked to this article.

## Data availability

The ATAC-seq and RNA-seq data that support the findings of this study have been deposited in NCBI BioProject (https://www.ncbi.nlm.nih.gov/bioproject/?term=PRJNA630597) with the accession code PRJNA630597. All other data are available from the corresponding author upon reasonable request. Source data are provided with this paper.

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

## Acknowledgements

We thank Lesley Devine and Chao Wang for cell sorting. We thank Mei Zhong from Yale Stem Cell Center Genomics Core facility for RNA-seq library preparation and sequencing. This work was supported by grants DK057846 and DK045735 from the NIH, grant and SRA2014-142-S-B from the Juvenile Diabetes Research Foundation, a Pilot award from the Yale Diabetes Research Center P30DK116577, and a gift from the Howalt family. The Yale Stem Cell Center Genomics Core was supported by the Connecticut Regenerative Medicine Research Fund and the Li Ka Shing Foundation.

## Author contributions

J.R. and K.C.H. conceived and designed the experiments and analyzed data. J.R. and S.D. conducted most of the cellular and animal experiments. A.L.P. acquired the human pancreatitis slides, prepared the MHCI tetramer for IGRP staining, and performed the human islets culture experiments. G.P. and M.L.-R. labeled and viewed the human pancreas slides with the supervision from D.P., R.K. and N.L. processed the samples for ATAC-seq and analyzed the data with the supervision from M.L.S. T.S. prepared RNA and cDNA amplification for RNA-seq libraries. M.L.S., D.P. and J.L. provided valuable inputs. J.R. and K.C.H. wrote the paper with input from all the authors.

## Competing interests

The authors declare no competing interests.
