## [Peer Review File · Nature Communications]

Reviewers' Comments:

Reviewer #1:

Remarks to the Author:

In the present study Rui and co-workers use relevant mouse models and human pancreas histology (plus a human islet experiment) to evaluate the role for Tet2 in immune-induced beta cell death. Based on the data obtained, the authors suggest that Tet2 is a regulator of the crosstalk between beta cells and the immune system, modulating intrinsic protective pathways in the beta cells. These conclusions are interesting, novel and relevant for a better understanding of the pathogenesis of type 1 diabetes. The mouse data are solid, but the human data less so (see detailed comments below), and the authors should perform additional experiments to support their hypothesis.

1. In Fig. 1a, islets from 6 wk B6 mice are not the ideal control for Tet2 expression in 12 w NOD mice. It would be important to also examine Tet2 expression in islets from NOD-Scid mice at 6 and 12 ws.
2. In Figures 1b and 2b it would be important to also evaluate whether IL-1, IL-6, IFN-gamma or IL-17-A alone induce Tet2 expression. Additionally, different cytokines are used in different experiments and no rationale is provided for this diverse choice of cytokines.
3. It is mentioned in Figure 2 that "...fold of Tet2 mRNA induced by cytokines relative to media was shown...". Media of whom? Control non-treated human islets? Furthermore, the data shown were obtained based on islets from a single organ donor, and should thus be considered as n=1. It is difficult to draw conclusions based on a single donor, and the authors must perform additional experiments in other preparations to confirm these findings. I suggest that they use these preliminary data (n=1) to make power calculation and, based on it, determine how many independent human islet preparations they should use to obtain reliable conclusions (from the data shown, it will be probably 3-4 preparations).
4. It is crucial to repeat the experiments shown in Figure 4 (performed in mouse islets) with human islet cells KD for Tet2 using specific siRNAs. The key question is whether KD of Tet2 in human beta cells will protect them against pro-inflammatory cytokines or, if feasible, cytotoxic T-cells.
5. In light of the author's previous findings, was there a change in PDL1 expression in the beta cells from the Tet2 KO mice?
6. The Pasquali's group has recently published ATAC-seq data from human islet cells exposed to cytokines (Nat Genet. 2019 Nov;51(11):1588-1595). It would be interesting to compare these findings with the mouse ATAC-seq data shown in Figure 7.
7. Actb was used as a housekeeping gene. Is the expression of this gene stable under the different experimental conditions used? This information should be provided in Methods.

Minor Comments

- On line 327 it should be "chemokines" and not "cytokines" (CXCL10 is mentioned below, on line 329), and supporting references should be provided for the statement that "...Release of inflammatory cytokines (or chemokines), which is well documented to occur from beta cells (refs?)..."
- The depth of sequencing used in the "low-input RNA-seq analysis" should be indicated.
- On lines 444 and 445 use "was" instead of "would then been" or "would be ready".
- The References in the text are provided with or without parenthesis. This should be made uniform in line with the journal instructions.

Reviewer #2:

Remarks to the Author:

In their manuscript, "Tet2 controls beta cells responses to inflammation in type I diabetes", Rui and co-authors study the impact of Tet2 on beta cell destruction mostly in mice, with very limited human samples included. As presented, there are limited data on the effects of Tet2 on the epigenetic profile of specific cell types, which limits mechanistic conclusions that the authors can make. Suggestions for significant revisions include:

1) How much do mouse models recapitulate what is seen in human type I diabetes? How much can you directly extrapolate given the small numbers of human samples studied?

2) In the Introduction, the authors make the statement that "Even 50 years after the diagnosis of T1D about 2/3 of patients still have detectable levels of C-peptide indicating residual beta cells". Maybe this is completely accepted with the diabetes field, but it seems plausible that other cell types in the body could produce C-peptide in the setting of autoimmunity...? Can the authors provide a reference for this statement? Likewise, the same question applies to their statement about levels of proinsulin in these individuals. Again, why can't other cell types produce these protein/protein fragments? Reference? Expression of MafA, PDX-1, and NeuroD in bone marrow-derived mesenchymal stromal cells (non-beta cells) can induce insulin production, for example.

3) In Figure 1, the authors examine Tet expression; what about Dnmt expression (the authors should remind readers here of their previous findings)? A review of very recent literature shows that TET3 is a direct HIF-1 target [Cao J et al, Blood Adv Jul 14 2020]. Could the authors be underestimating Tet3 expression levels if their studies were not conducted under hypoxic conditions? Along these lines, the experiments leading to Figure 1b were almost certainly carried out in normoxic conditions, and therefore, the lack of Tet3 induction would almost be predicted. These experiments should be repeated in hypoxia if at all possible to address the role of Tet3. If these experiments cannot be performed for some reason, then a discussion of these points should be included to inspire future work.

4) Figure 2 focuses on single patients: How can these results be generalized? Also, the results are not consistent between the presence of beta cells and TET2 expression so it is hard to draw any conclusions from these data. Therefore, I suggest that they be moved to the Supplementary data section and de-emphasized in a revision. In the Discussion, the authors describe these studies as if they studied multiple human subjects, which inaccurately reflects the work presented. If the authors can expand the number of human samples used, then such generalizations might be more justified.

5) In Tet2 deficient mice, progression to leukemia is dependent on translocation of gut microbiota that induce systemic inflammation characterized by IL-6: In the authors' colonies of Tet2-deficient mice, what is the microbiota status? Are their mice developing leukemia, and if so, what effect is the leukemia having on beta cell destruction (or lack thereof)? Although the experiments to address the role of the microbiota of these mice may be beyond the scope of this manuscript, a discussion about the possible role of the microbiota in mediating inflammation and development of diabetes is missing and could provide an important direction for future work.

6) Along these lines, the authors' transplantation protocol employed irradiation, which is known to break down the gut barrier. Therefore, it is possible that the gut microbiota play a role post-irradiation/transplantation and mediate resistance to beta cell death seen in the Tet2 KO mice. Did the authors measure inflammatory cytokines post-transplant? Translocation of gut bacteria? Again, more discussion of mechanism is appropriate.

7) Again, the experiments shown in Figure 4 were performed in normoxic conditions most likely. Can these experiments be repeated in hypoxia, which would more closely mimic those conditions found in vivo, and can RNA-seq/Nanostring be used to characterize gene expression changes with culture with inflammatory mediators? What happens to cell death under these conditions? Tet3 expression?

8) The authors study Tet2 deficiency, but make no measurements of 5-hydroxymethylcytosine (5-hmC) levels. Correlation of low levels of 5-hmC near open-chromatin regions can make a mechanistic connection between mouse genotype, changes to beta cell epigenome, altered gene expression profile, and beta cell function.

9) In the Discussion, the authors describe a paradoxical effect of Tet2 deficiency and decreased gene expression. This reflects a simplified understanding on the part of the authors. In every example well studied, decrease in expression of Tet enzyme(s) results in some increase locally in

5-hmC density, with overall lower levels. Gene expression follows consequently from local epigenetic modifications. Therefore, it is not a surprise at all that some genes show decreased expression in Tet2 deficiency. Some of these may also reflect indirect effects, for example, from increased expression of transcriptional repressors. The authors should make appropriate changes to this paragraph.

10) Have the gene expression/ATAC-seq data been deposited into publicly accessible databases? If so, the accession numbers should be included in the manuscript.

Minor issues:

1. The title is missing an apostrophe in 'cells' to indicate possession: The title should read: Tet2 controls beta cells' responses to inflammation in type I diabetes".
2. In line 51, data is a plural noun; sentence should read: ...data suggest...
3. In line 65, 'immune' is misspelled.
4. Line 153: a proper reference should be used, not just a weblink to a published paper.
4. Throughout the manuscript, gene names should be placed in italics when referring to nucleic acids; plain text should be used for proteins.

Reviewer #3:

Remarks to the Author:

The manuscript by Rui and colleagues presents evidence for the role of the Tet2 in regulating β -cell immunogenicity. Overall, the amount and quality of the data collected is impressive. The authors' careful statements are backed up with elegant experimental settings showing that the expression of Tet2 in β cells of NOD mice is (self)destructive.

My comments and questions are listed below (together with an approximate row number):

~117- results from the experiment in figure 1b were achieved from whole islets, it would be nice to know if all β cells respond with Tet2 upregulation or only a subset of cells- a flow cytometry analysis will be best since it will also allow to check for the granularity in the context of the authors previous results (Ref13).

~125- the images should be quantified and conclusions legitimized with statistics, for example: infiltrates and tet2 expression.

~157- As previous results from the authors(ref13) suggest, 2 subpopulations of β cells differ in granularity and differentiation features ('top' and 'bottom' subsets). It is shown here that bottom cells are Tet2 low. However, it is not clear whether the Tet2KO β cells have the features of the dedifferentiated cells. the authors could easily answer this by analyzing the differences in the transcriptomes of β cells from wt\Tet2 KO mice. In addition, in figure 1d it is clear that some cells of the diabetic are Tet2 positive while some are negative- are the positive cells SSC low, bottom cells?

~169-(fig3b) What is the reason for the huge variability in the of the percentage of Cd45+ cells in islets? Is it merely due to difference in the infiltration\migration process? Is it a lower islet cell number that cause over-representing of infiltrates? Is Tet2 expression heterogeneity inter or intra islet? ie. do some islets have more and others less? Additionally, the normalization chosen in the y axis may be problematic\misleading- the stable β cell (% of Cd45-) curve of the beta-KO condition could be a result of whole islet cell kill (unspecific), whereas in the beta-WT condition there is a targeted β cell elimination.

~196- I did not understand why NOD\scid mice were used here? The Tet2 KO mice are all on NOD background. Second it would be nice to see if the surviving cells from WT+infiltrates express Tet2 (an additional staining of Tett2 could have been added to the staining panel). The figure should show the existence of 2 populations (negative \ positive for Tet2) in the WT-media condition and one (Tet2 negative) in the WT+infiltrates condition.

~228- 'The T cells from the KO recipients showed reduced frequencies of CCR7+ and CXCR3+' that are proteins responsible for "homing"- how does this settle with the fact that 'the frequency of infiltrating cells was similar in the KO and WT islets (Supplementary Fig. 5)'?

~240- Did the authors check for antibody against MHCII?

~242- is there any sign for β cell de-differentiation as a defensive mechanism in the KO?

Do de-differentiated cells have low Tet2? It should be easy to check for it in existing datasets.

~254- The author note 'Tet2 deficiency does not lead to widespread chromatin remodeling of β cells', this is a well-placed comment and the result could be moved to the supplements. Or, the data could be reanalyzed more strictly, focusing on the top genes (that are up or down in all biological replicates). As for the downstream analysis, Tet2ko β cells have open chromatin in areas with motifs that are predicted to be bound by mature β cell TFs. however, the biological relevance of this particular result is not clear? What are the genes? and does the list match to data from the RNA-seq ?

Additional minor concern, the authors suggest that Tet2 deficiency results in opening and closing of a subset of islet TF binding sites. The data were generated from β cells of WT-NOD or Tet2 KO-NOD mice 8 weeks after transplantation of WT-NOD immunocytes. This result could be due to differences in the reaction to the immune cells but could also be native in the Tet2KO β cell. in case they have the data\material: It would be nice to see if there are fundamental, differences between the two genotypes, before being attacked by immunocytes, this will answer whether Tet2 KO β cells are primed, or only protected upon immune reaction.

~311 'the absence of TET2 expression suggests an association with survival': overall, I agree with this statement but I suggest to add a sentence about the possibility of the existence of cells that simply do not react to cytokines from immune cells.

Reviewer #4:

Remarks to the Author:

Rui et al showed that 1) Beta cells in both autoimmune mice and human upregulated expression of TET2 at disease onset, and the residual beta cells in diabetic mice and human down-regulated; 2) TET2 KO NOD mice transplanted with syngeneic NOD bone marrow had markedly reduced T1D frequency as compared with WT NOD recipients. The reduction of T1D in TET2 KO NOD mice was associated with resistance of beta cells as indicated by no reduction of beta cell percentage but not reduction of expansion of autoreactive CD8+ T cells in the spleen; 3) RNA-seq analysis showed that T cells from TET2 KO islets have down-regulated expression of pathogenic phenotype, and flow cytometry analysis showed that those T cells have reduced percentage of CD4+ or CD8+ CXCR3+ T cells; 4) TET2 KO beta cell but not the alpha cells are more resistant to cytokine or islet infiltrating cells induced apoptosis; 5) RNA-seq analysis showed that TET2 KO beta cells have reduced expression of inflammatory response genes; and ATAC-seq analysis showed that TET2 KO beta cells have reduced chromatin accessibility at inflammatory mediator binding sites. The authors conclude that TET2 regulate pathologic interactions between immune cells and beta cells and control intrinsic protective pathways. These are interesting observations. However, the conclusion is not supported by mechanistic details.

1. Can overexpression of TET2 in beta cells augment beta cell susceptibility towards inflammatory cytokines or augment beta cell death or augment T cell activation?
2. In Figure 3, the insulinitis status of TET2 KO mice at 20 weeks after bone marrow transplantation need to be evaluated.
3. In Figure 3, the percentage and yield as well as activation/anergy status of IGRP+CD8+ T cells should be evaluated to indicate the impact of TET2KO beta cells on autoreactive T cells.
4. In Figure 3, the percentage of Foxp3+ Treg cells and activation status in the pancreatic lymph nodes and pancreas should be evaluated, as it was reported by different groups that Treg cells in the pancreatic lymph node and pancreas play a critical role in T1D pathogenesis in NOD mice

(Tang, Q. et al Immunity; Zhang, M et al: PNAS 2018:
www.pnas.org/cgi/doi/10.1073/pnas.1720169115).

5. Figure 4, 6 and 7 should be linked together to form a cohesive story. A functional link needs to be shown.

6. In Figure 5, RNA-seq analysis indicates that beta cell expression of TET2 regulates interaction between beta cells and T cells. The indication needs to be validated with functional experiments. For example, how beta cell TET2 KO leads to reduction of CXCR3 expression in T cells.

7. The statement "Immune responses are not activated by TET2 deficient islet cells" is overstated.

Responses to Reviewers: NCOMMS-20-21896 “Tet2 Controls the Responses of β cells to Inflammation in Type 1 Diabetes

Reviewer #1 (Remarks to the Author):

In the present study Rui and co-workers use relevant mouse models and human pancreas histology (plus a human islet experiment) to evaluate the role for Tet2 in immune-induced beta cell death. Based on the data obtained, the authors suggest that Tet2 is a regulator of the crosstalk between beta cells and the immune system, modulating intrinsic protective pathways in the beta cells. These conclusions are interesting, novel and relevant for a better understanding of the pathogenesis of type 1 diabetes. The mouse data are solid, but the human data less so (see detailed comments below), and the authors should perform additional experiments to support their hypothesis.

We appreciate the Reviewer's supportive comments.

1. In Fig. 1a, islets from 6 wk B6 mice are not the ideal control for Tet2 expression in 12 w NOD mice. It would be important to also examine Tet2 expression in islets from NOD-Scid mice at 6 and 12 ws.

As suggested by the reviewer, we took 6wk B6 out and add in Tet2 expression in islets from NOD/*scid* mice at 5wks and 11wks to age match the NOD mice. See in revised Fig.1a.

2. In Figures 1b and 2b it would be important to also evaluate whether IL-1, IL-6, IFN-gamma or IL-17-A alone induce Tet2 expression. Additionally, different cytokines are used in different experiments and no rationale is provided for this diverse choice of cytokines.

We agree. In the revised version, we examined individual cytokine as well as cytokine combinations on *Tet2* induction both in mouse and in human islets. See in revised Fig. 1c and 2b.

Based on these findings and for consistency, we show the effects of the cytokine combination of $\text{TNF}\alpha + \text{IL-1}\beta + \text{IFN}\gamma$, which are the most well documented cytokines in the pathogenesis of T1D in our studies of Tet2 in mouse and TET2 in human islets.

3. It is mentioned in Figure 2 that “...fold of Tet2 mRNA induced by cytokines relative to media was shown...” Media of whom? Control non-treated human islets? Furthermore, the data shown were obtained based on islets from a single organ donor, and should thus be considered as n=1. It is difficult to draw conclusions based on a single donor, and the authors must perform additional experiments in other preparations to confirm these findings. I suggest that they use these preliminary data (n=1) to make power calculation and, based on it, determine how many independent human islet preparations they should use to obtain reliable conclusions (from the data shown, it will be probably 3-4 preparations).

We clarified in the legends to Fig. 2 that the fold is relative to non-treated human islets cultured in media alone. We added 3 more human donors to the previous 2 human donors. Therefore, the data in Fig. 2c are from 5 donors.

4. It is crucial to repeat the experiments shown in Figure 4 (performed in mouse islets) with human islet cells KD for Tet2 using specific siRNAs. The key question is whether KD of Tet2 in human beta cells will protect them against pro-inflammatory cytokines or, if feasible, cytotoxic T-cells.

We agree with the reviewer and tried commercially available specific siRNA (smartpool from Accell) to knock down TET2 in human islets. The TET2 knock down was not reliable, most likely due to the culture requirements for human islets and those needed for efficient knock-down with siRNA. Alternatively, we used R-2HG which is frequently employed to inhibit the activity of TET2^{1,2}. When cultured with cytokine cocktail, we found reduced cytokines and chemokines production as well as the cell death receptors in human islets as seen in Tet2 KO mouse islets. See new data in Fig. 6d.

5. In light of the author's previous findings, was there a change in PDL1 expression in the beta cells from the Tet2 KO mice?

To determine the effects of Tet2 on PD-L1 expression on islet cells, we analyzed expression of PD-L1 and other relevant molecular during culture with IFN γ . We used B6 WT and KO islets for these studies because we wanted to analyze the changes in the absence of inflammatory cells. This updated analysis is shown in Supplementary Fig 8, in which *Pd1l* as well as *Cxcl10* and *Cxcl11* expression are lower in β cells from Tet2 KO vs WT B6 mice

6. The Pasquali's group has recently published ATAC-seq data from human islet cells exposed to cytokines (Nat Genet. 2019 Nov;51(11):1588-1595). It would be interesting to compare these findings with the mouse ATAC-seq data shown in Figure 7.

We feel that these an ATAC-seq analysis of human islets with and without cytokines is beyond the scope this paper. We agree that this is an important experiment to do and plan to do it in the future.

7. *Actb* was used as a housekeeping gene. Is the expression of this gene stable under the different experimental conditions used? This information should be provided in Methods.

We have done studies with *Gapdh* and *Hprt* as housekeeping genes as well and found similar results, and thus settled with *Actb* alone as housekeeping gene. The *Actb* doesn't change significantly under different experimental conditions.

Minor Comments

- On line 327 it should be "chemokines" and not "cytokines" (CXCL10 is mentioned below, on line 329), and supporting references should be provided for the statement that "...Release of

inflammatory cytokines (or chemokines), which is well documented to occur from beta cells (refs?)...”

We corrected the mistake and added in the relevant references.

- The depth of sequencing used in the “low-input RNA-seq analysis” should be indicated.

We indicated the depth of the sequencing in the method.

- On lines 444 and 445 use “was” instead of “would then been” or “would be ready”.

We corrected the grammar.

- The References in the text are provided with or without parenthesis. This should be made uniform in line with the journal instructions.

We made the changes that are in line with the journal instructions.

Reviewer #2 (Remarks to the Author):

In their manuscript, “Tet2 controls beta cells responses to inflammation in type I diabetes”, Rui and co-authors study the impact of Tet2 on beta cell destruction mostly in mice, with very limited human samples included. As presented, there are limited data on the effects of Tet2 on the epigenetic profile of specific cell types, which limits mechanistic conclusions that the authors can make. Suggestions for significant revisions include:

We appreciate the Reviewer’s suggestions. In response, we have added additional human samples and clarified the number of individual studies, and also present data on the effects of Tet2KO on α cells in Fig 3f. Moreover, in our new analysis, shown in Fig. 6c, we identify the epigenetic changes in specific genes in the KO β cells. The other changes are described below in the response to queries.

1) How much do mouse models recapitulate what is seen in human type I diabetes? How much can you directly extrapolate given the small numbers of human samples studied?

We agree with the concerns of the Reviewer about the limitations of NOD studies and mouse studies in general.. Despite the limitations, for the past 4 decades, the NOD mouse has been the primary animal model for studying autoimmune diabetes, and has shed light on human pathologic processes in immune and β cells as well as identifying therapies that have proven to be beneficial to humans³. Murine islet studies have provided a great deal of information about human islets even through there are several recognized differences. Particularly for studying the interactions between immune and β cells a better or even another suitable animal model or in vitro culture system does not exist. Nonetheless, we agree that having more robust human data would be important and therefore we have increased the number of samples that were taken from individual human donors in our studies of human islets in Fig. 2. We now show TET2 data induced by cytokines in 5 different islet donors in revised Fig. 2c.

2) In the Introduction, the authors make the statement that “Even 50 years after the diagnosis of T1D about 2/3 of patients still have detectable levels of C-peptide indicating residual beta cells”. Maybe this is completely accepted with the diabetes field, but it seems plausible that other cell types in the body could produce C-peptide in the setting of autoimmunity...? Can the authors provide a reference for this statement? Likewise, the same question applies to their statement about levels of proinsulin in these individuals. Again, why can’t other cell types produce these protein/protein fragments? Reference? Expression of MafA, PDX-1, and NeuroD in bone marrow-derived mesenchymal stromal cells (non-beta cells) can induce insulin production, for example.

The reference is Keenan HA et al Diabetes 59:2846, 2010. To our knowledge, β cells are the only cells capable of insulin production and release even in the setting of autoimmunity. Other cells do have low levels of gene transcription but these cells lack the enzymes (PC 1,3) needed for processing of proinsulin.

We have modified the sentences in the Introduction to: Even 50 years after the diagnosis of T1D about 2/3 of patients still have detectable levels of C-peptide indicating residual β cells ⁴. Moreover, the increased relative levels of proinsulin suggests dysfunctional but persistent β cells ⁵. The reasons why some β cells survive and others succumb to autoimmune killing are not certain.

3) In Figure 1, the authors examine Tet expression; what about Dnmt expression (the authors should remind readers here of their previous findings)? A review of very recent literature shows that TET3 is a direct HIF-1 target [Cao J et al, Blood Adv Jul 14 2020]. Could the authors be underestimating Tet3 expression levels if their studies were not conducted under hypoxic conditions? Along these lines, the experiments leading to Figure 1b were almost certainly carried out in normoxic conditions, and therefore, the lack of Tet3 induction would almost be predicted. These experiments should be repeated in hypoxia if at all possible to address the role of Tet3. If these experiments cannot be performed for some reason, then a discussion of these points should be included to inspire future work.

We address the previous work. “In our previous studies we found epigenetic changes in β cells from NOD mice that led to methylation marks in *Ins1* and *Ins2* and reduced gene transcription and similar changes in INS when human islets were cultured with inflammatory cytokines.”

We agree with the Reviewer’s point and meanwhile we would like to point out that insulin-producing pancreatic β cells are highly susceptible to oxygen deficiency ^{6,7}. We do not have any data to suggest that Tet3 is involved in the adaptation of β cells to immunologic stress. As shown in Fig. 1, the levels of Tet3 did not increase during diabetes progression in NOD mice. It is possible that under hypoxic conditions, which may occur for example during transplantation, TET3 expression is increased, but we do not have data to support this or to direct investigations of TET3 in islets during autoimmune diabetes. Tet2 and Tet3 have recently been described as “guardians” of Treg stability and immune homeostasis but it is important to note that in our bone marrow transplant studies the immune cells have normal expression of Tet2 and 3 ⁸.

4) Figure 2 focuses on single patients: How can these results be generalized? Also, the results are not consistent between the presence of beta cells and TET2 expression so it is hard to draw any conclusions from these data. Therefore, I suggest that they be moved to the Supplementary data section and de-emphasized in a revision. In the Discussion, the authors describe these studies as if they studied multiple human subjects, which inaccurately reflects the work presented. If the authors can expand the number of human samples used, then such generalizations might be more justified.

We may not have clearly indicated the number of individuals represented by the data in Fig. 2 in our previous version. In order to present the histology, we show representative individuals but as we indicated in the legend Fig. 2: “Data represent normal individuals (n=8), donors with autoimmune pancreatitis (n=5), non-diabetic donors who were autoantibody+ (n=7), and C-peptide+ patients with T1D of relatively short duration (n=7).”

We agree with the need to evaluate multiple donors and to do so in a quantitative manner. In the revised version, we employed Image J to quantify the fluorescence intensity of TET2 staining in different conditions. We picked 20 islets per condition and compared different conditions, each condition is from 3-7 donors. These data are shown in Fig. 2b in the revised version.

Also in the legend to Fig. 2: “Data (mean \pm SEM) are from 3 experiments, each with 2,000 islet equivalency (IEQ) from one non-diabetic donor.” In the revised version, we included 2 extra donors to repeat the experiment Fig2b so the total number of different human islet donors is now 5.

5) In Tet2 deficient mice, progression to leukemia is dependent on translocation of gut microbiota that induce systemic inflammation characterized by IL-6: In the authors’ colonies of Tet2-deficient mice, what is the microbiota status? Are their mice developing leukemia, and if so, what effect is the leukemia having on beta cell destruction (or lack thereof)? Although the experiments to address the role of the microbiota of these mice may be beyond the scope of this manuscript, a discussion about the possible role of the microbiota in mediating inflammation and development of diabetes is missing and could provide an important direction for future work.

We agree that an analysis of the microbiome of the mice is beyond the scope of this manuscript. Our KO NOD or B6 mice didn’t develop leukemia during 36 weeks of follow up but this was not the focus of our investigations. We have added the suggested discussion of the microbiome and included a reference pertaining to this interaction (Ansari et al Nat Microbiol 2020).

6) Along these lines, the authors’ transplantation protocol employed irradiation, which is known to break down the gut barrier. Therefore, it is possible that the gut microbiota play a role post-irradiation/transplantation and mediate resistance to beta cell death seen in the Tet2 KO mice. Did the authors measure inflammatory cytokines post-transplant? Translocation of gut bacteria? Again, more discussion of mechanism is appropriate.

Please see our response above. We note that both the Tet2 KO and WT mice received the same bone marrow transplant with the same procedures including the irradiation.

7) Again, the experiments shown in Figure 4 were performed in normoxic conditions most likely. Can these experiments be repeated in hypoxia, which would more closely mimic those conditions found in vivo, and can RNA-seq/Nanostring be used to characterize gene expression changes with culture with inflammatory mediators? What happens to cell death under these conditions? Tet3 expression?

Please see answers to point 3 above. It is not clear what hypoxic conditions would be appropriate. Our point was to study the role of Tet2 and not hypoxia on β cell injury and responses.

8) The authors study Tet2 deficiency, but make no measurements of 5-hydroxymethylcytosine (5-hmC) levels. Correlation of low levels of 5-hmC near open-chromatin regions can make a mechanistic connection between mouse genotype, changes to beta cell epigenome, altered gene expression profile, and beta cell function.

We agree with the reviewer and performed hydroxymethylation DNA Immunoprecipitation to use specific purified 5-hmC antibody to immunoprecipitate and enrich for DNA fragments containing 5-hydroxymethylcytosine (5-hmC) from β cells that are either from KO or WT recipients of BMT.

QPCR was performed paired with primers that are specific for promoter regions of genes of interest where CpG sites are present within the 500bp of the transcription start site (TSS). We found, in agreement with the in vitro data showing reduced level of chemokines and cytokines and cell death receptors in KO islets when cultured with pro-inflammatory cytokine cocktails (Fig. 6a,b), the 5-hmC level in the promoter regions of these genes dropped significantly in KO vs WT. These new studies are shown in Fig. 6c.

9) In the Discussion, the authors describe a paradoxical effect of Tet2 deficiency and decreased gene expression. This reflects a simplified understanding on the part of the authors. In every example well studied, decrease in expression of Tet enzyme(s) results in some increase locally in 5-hmC density, with overall lower levels. Gene expression follows consequently from local epigenetic modifications. Therefore, it is not a surprise at all that some genes show decreased expression in Tet2 deficiency. Some of these may also reflect indirect effects, for example, from increased expression of transcriptional repressors. The authors should make appropriate changes to this paragraph

We agree and have made changes to the paragraph wording and a new reference (Cull et al).

10) Have the gene expression/ATAC-seq data been deposited into publicly accessible databases? If so, the accession numbers should be included in the manuscript.

Yes and the accession number was provided at submission under the section of Data Availability

Minor issues:

1. The title is missing an apostrophe in ‘cells’ to indicate possession: The title should read: Tet2 controls beta cells’ responses to inflammation in type I diabetes”.

We modified the title to: “Tet2 Controls the Responses of β cells to Inflammation in Type 1 Diabetes” which we believe is grammatically improved and more precise.

2. In line 51, data is a plural noun; sentence should read: ...data suggest...

We made the change as suggested and have made modifications throughout the manuscript.

3. In line 65, ‘immune’ is misspelled.

This has been corrected.

4. Line 153: a proper reference should be used, not just a weblink to a published paper.

We updated the references in this section.

4. Throughout the manuscript, gene names should be placed in italics when referring to nucleic acids; plain text should be used for proteins.

We made the changes as suggested

Reviewer #3 (Remarks to the Author):

The manuscript by Rui and colleagues presents evidence for the role of the Tet2 in regulating β -cell immunogenicity. Overall, the amount and quality of the data collected is impressive. The authors' careful statements are backed up with elegant experimental settings showing that the expression of Tet2 in β cells of NOD mice is (self) destructive.

We appreciate the Reviewer's supportive comments and constructive suggestions.

My comments and questions are listed below (together with an approximate row number):

~117- results from the experiment in figure 1b were achieved from whole islets, it would be nice to know if all β cells respond with Tet2 upregulation or only a subset of cells- a flow cytometry analysis will be best since it will also allow to check for the granularity in the context of the authors' previous results (Ref13).

In that previous study, we described subpopulations of β cells, designated as "top" and "bottom" based on high and low granularity, with different susceptibilities to killing. In Supplementary Figure 4, we identify differences in Tet2 expression in these β cell subpopulations which is consistent with resistance to killing with decreased expression of Tet2.

~125- the images should be quantified and conclusions legitimized with statistics, for example: infiltrates and tet2 expression.

We agree with the reviewer and used Image J to quantify the fluorescence intensity of TET2 staining in different conditions. We picked 20 islets per condition and compared different conditions. For each condition tissues from 3-7 donors were analyzed. This analysis is shown in Fig. 2b. We did not include infiltrates for quantification as infiltrates were not found in all conditions or in all cases or islets from the same condition.

~157- As previous results from the authors (ref13) suggest, 2 subpopulations of β cells differ in granularity and differentiation features ('top' and 'bottom' subsets). It is shown here that bottom cells are Tet2 low. However, it is not clear whether the Tet2KO β cells have the features of the dedifferentiated cells. the authors could easily answer this by analyzing the differences in the transcriptomes of β cells from wt/Tet2 KO mice. In addition, in figure 1d it is clear that some cells of the diabetic are Tet2 positive while some are negative- are the positive cells SSC low, bottom cells?

Tet2KO β cells did not have gene signatures of dedifferentiation. We analyzed the composition of the islet cells (β cells, α cells and the other islet cells) as well as the profile of the key transcription factors and genes that define β cells. We did not identify differences between the Tet2KO vs WT beta cells in the genes that we had identified as differentially expressed in the dedifferentiated β cells such as *Gcg* and *Ins*. Please see Supplementary Figure 1c-1e. Our interpretation of the data in Fig. 1d is not of two discrete populations which is different from what we had seen in our studies of β cells from WT NOD mice.

~169-(fig3b) What is the reason for the huge variability in the of the percentage of Cd45+ cells in islets? Is it merely due to difference in the infiltration\migration process? Is it a lower islet cell number that cause over-representing of infiltrates?

The islets were harvested from mice from 8-20 weeks after the transplant and therefore a range of infiltrates would be expected. We have clarified this in the text. Many investigators have described heterogeneity in islet infiltrates between mice and even between islets from the same pancreas and therefore this data was not surprising.

Is Tet2 expression heterogeneity inter or intra islet? ie. do some islets have more and others less?

Our new analysis of human islets shown in Fig 2b describes the variability of TET2 expression between islets but we are unable to analyze all of the cells within an islet to address this question precisely. Among the 20 islets that were examined, the CV was 49.5% in the pancreases from patients with T1D.

Additionally, the normalization chosen in the y axis may be problematic\misleading- the stable β cell (% of Cd45-) curve of the beta-KO condition could be a result of whole islet cell kill (unspecific), whereas in the beta-WT condition there is a targeted β cell elimination.

In Fig3b, in the y axis, remaining beta cells is presented as the percentage of CD45 negative cells of all of the cells isolated from the islets. The x axis indicates the infiltration level which reflects the percentage of the CD45+ cells of the total islet cells. The islets that were used for the analysis were handpicked and so therefore, all had CD45- cells. Of course, we only know about the islets that we can pick - the islets in which the cells were all killed were not analyzed in these experiments. Nonetheless, by comparing the proportion of β cells among the CD45- cells, we have a measure of the β cell specific killing. We generally are able to isolate around 100 islets per mouse from the KO recipients. We are able to isolate fewer islets from the WT recipients since the number of recoverable islets decreases as the disease progress. This is shown below (Figure 1).

Figure 1: Recoverable islets from KO BM transplant recipients or WT NOD mice. Data on the L (KO mice) were from the mice used in the studies shown in Fig 3b. In the WT NOD mice, in whom diabetes develops over time, the number of recoverable islets decreases. In all of our WT BMT recipients, diabetes development was rapid and there were small numbers of recoverable islets.

~196- I did not understand why NOD/scid mice were used here? The Tet2 KO mice are all on NOD background. Second it would be nice to see if the surviving cells from WT+infiltrates express Tet2 (an additional staining of Tet2 could have been added to the staining panel). The figure should show the existence of 2 populations (negative \ positive for Tet2) in the WT-media condition and one (Tet2 negative) in the WT+infiltrates condition.

NOD/scid mice lack functional T cells and B cells. In Fig.4b, beta cells from NOD/scid mice were used in the co-culture experiment with CD45+ infiltrates from WT NOD mice to examine the β cell death that occurs with co-culture with infiltrates. We used NOD/scid to avoid any killing or damage that might have initiated already in NOD mice when the islets are harvested since these infiltrates can be present as early as 3 weeks of age.

The purpose of this experiment was to look at cell survival. The experiment shows on the same genetic basis the role of Tet2 and that β cells lacking Tet2 are protected from killing either from infiltrates or pro-inflammatory cytokines.

~228- ‘The T cells from the KO recipients showed reduced frequencies of CCR7+ and CXCR3+’ that are proteins responsible for “homing”- how does this settle with the fact that ‘the frequency of infiltrating cells was similar in the KO and WT islets (Supplementary Fig. 5)’?

We analyzed the total immune cell composition in pancreatic lymph nodes. The flow data show that despite a similar frequency of infiltrating cells (CD45+) was similar between KO and WT recipients of WT bone marrow, the overall B cell frequency is slightly higher while T cells are of lower frequency in KO recipients. Thus, our data in Fig. 3b shows that the cells still infiltrate the islets. We have quantified the insulinitis in the KO recipient mice as follows:

Figure 2: Insulinitis score in 21-week Tet2 KO female NOD mice. (a) Representative pancreas slides showing the insulinitis score assignment. 0: no infiltration. 1: peri-insulinitis. 2: infiltrative insulinitis <50% of the islet. 3: infiltrative insulinitis >50% of the islet. (b) Shown here is the insulinitis score of a total of 112 islets. Data are collected from 30-40 islets from the pancreas of every mice (three mice in every condition). Every islet was analyzed blindly and every islet was given a score between 0 and 3. 48 islets are score 0(43%), 25 are score1 (22%), 26 are score 2(23%), 13 are score 3(17%).

It is clear that these mice had inflammatory cells in the islets as shown in Fig. 3b. We added text describing these findings. In addition, we phenotyped the cells and show this new data in Fig. 4b.

~240- Did the authors check for antibody against MHCII?

We checked by FACs the MHCII level on beta cells from KO vs WT (B6 background) mice and didn't identify any baseline level difference on CD45+ or islet cells. We did not find differences in the expression of Class I MHC molecules (Supplementary Figure 7) in WT and KO B6 mice, which are most relevant for killing by CD8+ T cells.

~242- is there any sign for β cell de-differentiation as a defensive mechanism in the KO

We did not identify transcriptional signs of β cell de-differentiation in the KO recipients of the WT bone marrow (see Fig. 5) and discussion above.

Do de-differentiated cells have low Tet2? It should be easy to check for it in existing datasets.

Yes, we show this analysis, based on studies from our previously publication in Supplementary Fig 4.

~254- The author note 'Tet2 deficiency does not lead to widespread chromatin remodeling of β cells', this is a well-placed comment and the result could be moved to the supplements. Or, the data could be reanalyzed more strictly, focusing on the top genes (that are up or down in all biological replicates). As for the downstream analysis, Tet2ko β cells have open chromatin in areas with motifs that are predicted to be bound by mature β cell TFs. however, the biological

relevance of this particular result is not clear? What are the genes? and does the list match to data from the RNA-seq ?

We appreciate the Reviewer's suggestion and have done additional analyses to address them. We agree it is important to show how Tet2 deficiency changes the epigenetic landscape since that is a major role of the enzyme. Rather than moving the data to the Supplement, we have merged Fig. 5 and Fig. 7 into Fig.5abcd in the revision. Our ATAC-seq analysis showed that Tet2 KO β cells have reduced chromatin accessibility at inflammatory mediators binding sites (Fig. 5d). As a result of that, the downstream chemokines and cytokines involved in the β cell immune response were reduced (Fig. 6ab) in mice. Consistent with these findings, we show that the hydroxymethylation level of these immune response genes were also reduced in KO β cells in the new data in Fig. 6c. Similar changes were also seen in human islets cultured with cytokines and TET inhibitor (Fig. 6d).

Additional minor concern, the authors suggest that Tet2 deficiency results in opening and closing of a subset of islet TF binding sites. The data were generated from β cells of WT-NOD or Tet2 KO-NOD mice 8 weeks after transplantation of WT-NOD immunocytes. This result could be due to differences in the reaction to the immune cells but could also be native in the Tet2KO β cell. in case they have the data\material: It would be nice to see if there are fundamental, differences between the two genotypes, before being attacked by immunocytes, this will answer whether Tet2 KO β cells are primed, or only protected upon immune reaction.

This is a good point. Please see Supplementary Figure 1 where comparisons were made between Tet2KO vs WT islets as well as β cells from B6 mice, which is without islet immune responses. We did not observe differences in morphology or function (glucose levels)

~311 'the absence of TET2 expression suggests an association with survival': overall, I agree with this statement but I suggest to add a sentence about the possibility of the existence of cells that simply do not react to cytokines from immune cells.

This is a good suggestion and we have revised the text. -to add a sentence about the possibility of the existence of cells that simply do not react to cytokines from immune cells.

Reviewer #4 (Remarks to the Author):

Rui et al showed that 1) Beta cells in both autoimmune mice and human upregulated expression of TET2 at disease onset, and the residual beta cells in diabetic mice and human down-regulated; 2) TET2 KO NOD mice transplanted with syngeneic NOD bone marrow had markedly reduced T1D frequency as compared with WT NOD recipients. The reduction of T1D in TET2 KO NOD mice was associated with resistance of beta cells as indicated by no reduction of beta cell percentage but not reduction of expansion of autoreactive CD8⁺ T cells in the spleen; 3) RNA-seq analysis showed that T cells from TET2 KO islets have down-regulated expression of pathogenic phenotype, and flow cytometry analysis showed that those T cells have reduced percentage of CD4⁺ or CD8⁺ CXCR3⁺ T cells; 4) TET2 KO beta cell but not the alpha cells are more resistant to cytokine or islet infiltrating cells induced apoptosis; 5) RNA-seq analysis showed that TET2 KO beta cells have reduced expression of inflammatory response genes; and ATAC-seq analysis showed that TET2 KO beta cells have reduced chromatin accessibility at inflammatory mediator binding sites. The authors conclude that TET2 regulate pathologic interactions between immune cells and beta cells and control intrinsic protective pathways. These are interesting observations. However, the conclusion is not supported by mechanistic details.

We agree with the Reviewer's summary. In response to the critique, we have done additional studies and revisions to address his/her concern about the support of the conclusions. We believe that with these new data, which includes studies with a TET2 inhibitor in human islet cells, the conclusions are strongly supported.

1. Can overexpression of TET2 in beta cells augment beta cell susceptibility towards inflammatory cytokines or augment beta cell death or augment T cell activation?

Previous work suggests that overexpression of Tet2 is associated with increased β cell killing⁹. Likewise in tumor cell lines increased activity of Tet2, with Vitamin C, can enhance killing¹⁰.

We do not have a reliable way to overexpress Tet2 in primary islet cells short of preparing a new transgenic model. This approach would also have limitations since to recreate the situation in the transplant studies, we would have to have conditional expression of the Tet2 at a time after maturation of the immune response and the islets. Whether overexpression did or did not enhance β cell killing, we did not feel that we would gain any additional insights into the mechanisms that indicate how loss of Tet2 protects β cells. While it would be possible to add Vitamin C to cultures of islets and immune cells any activity we would observe could be on the immune cells or the β cells and the effects of the vitamin on the cells would not be specific.

2. In Figure 3, the insulinitis status of TET2 KO mice at 20 weeks after bone marrow transplantation need to be evaluated.

Insulinitis score in 21-week Tet2 KO female NOD mice. Insulinitis score. 0: no infiltration. 1: peri-insulinitis. 2: infiltrative insulinitis <50% of the islet. 3: infiltrative insulinitis >50% of the islet. Data are collected from 30-40 islets from the pancreas of every mice (three mice in every condition).

Every islet was analyzed blindly and every islet was given a score between 0 and 3. We show these data above in our response to Reviewer #3 and have added text.

3. In Figure 3, the percentage and yield as well as activation/energy status of IGRP+CD8+ T cells should be evaluated to indicate the impact of TET2KO beta cells on autoreactive T cells.

We agree and have added in the new data showing the activation status of nrp7-tetramer+CD8T cells from WT and KO recipients in Fig. 4d.

4. In Figure 3, the percentage of Foxp3+ Treg cells and activation status in the pancreatic lymph nodes and pancreas should be evaluated, as it was reported by different groups that Treg cells in the pancreatic lymph node and pancreas play a critical role in T1D pathogenesis in NOD mice (Tang, Q. et al Immunity; Zhang, M et al: PNAS 2018: www.pnas.org/cgi/doi/10.1073/pnas.1720169115).

We analyzed the immune cell composition from pancreatic lymph nodes. We did not find significant differences in the percentage of Tregs (CD4+CD25+CD127low) between KO and WT recipients of WT bone marrow. In addition, analysis of Tregs is presented in the new Fig 4c. Therefore we interpret our data not to suggest that Tregs are responsible for the reduced killing of β cells.

Furthermore, Fig. 3d showing that immune cells with pathologic capability developed in the KO recipients since they cause disease in SCID recipients.

- 5. Figure 4, 6 and 7 should be linked together to form a cohesive story. A functional link needs to be shown.

We agree with this point. We performed hydroxymethylation DNA Immunoprecipitation to use specific purified 5-hmC antibody to immunoprecipitate and enrich for DNA fragments containing 5-hydroxymethylcytosine (5-hmC) from β cells that are either from KO or WT recipients of BMT. QPCR was performed paired with primers that are specific for promoter regions of genes of interest where CpG sites are present within the 500bp of TSS. We found that in agreement with the in vitro data showing reduced level of chemokines and cytokines as well as cell death receptors in KO vs WT islets when cultured with pro-inflammatory cytokine cocktails (Fig. 6a,b), the 5-hmC level in the promoter regions of these genes dropped significantly in KO vs WT (see new data Fig. 6c).

In addition, Supplemental Fig 8 shows that the expression of PD-L1, CXCL10, and CXCL11 are reduced in the KO islets when they are cultured with IFN γ . This is consistent with the ATACseq findings (Fig.5d) in which we identified reduced chromatin accessibility of the STAT1 locus. These findings are also consistent with what has been reported in studies of tumors¹⁰ (reference 19 in the manuscript). Therefore, we believe that this analysis adds to the coherent story that Tet2 elimination reduces the activation of inflammatory genes that are required for β cell killing.

6. In Figure 5, RNA-seq analysis indicates that beta cell expression of TET2 regulates interaction between beta cells and T cells. The indication needs to be validated with functional experiments. For example, how beta cell TET2 KO leads to reduction of CXCR3 expression in T cells.

We added in the new data to directly address this question about the mechanism of protection from diabetes. We show the activation status of NRPV7-tetramer+CD8T cells from WT and KO recipients in Fig4c. CD44 and PD1 level on tetramer+ CD8T cells are similar between WT and KO recipients. However, CXCR3 levels were significantly lower on the KO vs WT T cells, consistent with the reduced CXCR3+CD4/CD8T cells in Fig4b. *Cxcl10* is reduced in mouse islets cultured with inflammatory cytokines and *CXCL10* is also reduced in human islets cultured with inflammatory cytokines plus Tet2 inhibitor. Based on the well-defined role of the CXCL10/CXCR3 system in Type 1 Diabetes, we speculate that Tet2 KO leads to the reduced 5-hmC level of *Cxcl10* promoter region and the drop of *Cxcl10* induction, which is part of the inflammatory response from beta cells and which in turn is responsible for reduced expression of CXCR3 in T cells.

We agree with the Reviewer's and Editor's concern about the incomplete nature of a mechanism in the previous version of the manuscript. We believe our new data and analyses provide a clear mechanism that we feel is unique since it is not dependent on the specificity of T cells per se or even their intrinsic ability to respond to diabetes antigens. Rather it depends on the ability of β cells to provide the signal to T cells that enable them to perpetuate a "feed forward" loop that results in β cell killing. This functional cooperation between β cells and immune cells has not previously been identified or appreciated. We have added the following text to the Discussion to explain this mechanism: **Groom et al** described a CXCR3 inflammatory loop in which the IFN γ -induced chemokines recruit CXCR3 expressing T cells to inflammatory sites. The production of IFN γ by the CXCR3+ T cells can provide a "feed forward" in which additional CXCR3-expressing inflammatory cells, such as those thought to mediate autoimmune diabetes are recruited. Thus, the inability to respond to IFN γ may block this loop and result in the reduced frequency of the pathologic T cells^{11-14 15}. A unique feature of this mechanism of protection is that it does not depend on the specificity or the ability of the immune cells to respond to the target β cells per se but rather to the ability of the β cells to respond to inflammatory mediators and recruit and activate the T cells.

7. The statement "Immune responses are not activated by TET2 deficient islet cells" is overstated.

We have modified the text to be consistent with the data we present.

References

- 1 Ye, D., Ma, S., Xiong, Y. & Guan, K. L. R-2-hydroxyglutarate as the key effector of IDH mutations promoting oncogenesis. *Cancer cell* **23**, 274-276, doi:10.1016/j.ccr.2013.03.005 (2013).
- 2 Losman, J. A. *et al.* (R)-2-hydroxyglutarate is sufficient to promote leukemogenesis and its effects are reversible. *Science* **339**, 1621-1625, doi:10.1126/science.1231677 (2013).
- 3 Shoda, L. K. *et al.* A comprehensive review of interventions in the NOD mouse and implications for translation. *Immunity* **23**, 115-126, doi:10.1016/j.immuni.2005.08.002 (2005).
- 4 Keenan, H. A. *et al.* Residual insulin production and pancreatic β -cell turnover after 50 years of diabetes: Joslin Medalist Study. *Diabetes* **59**, 2846-2853, doi:10.2337/db10-0676 (2010).
- 5 Neyman, A. *et al.* Persistent elevations in circulating INS DNA among subjects with longstanding type 1 diabetes. *Diabetes Obes Metab* **21**, 95-102, doi:10.1111/dom.13489 (2019).
- 6 Komatsu, H., Kandeel, F. & Mullen, Y. Impact of Oxygen on Pancreatic Islet Survival. *Pancreas* **47**, 533-543, doi:10.1097/MPA.0000000000001050 (2018).
- 7 Zheng, X. *et al.* Acute hypoxia induces apoptosis of pancreatic β -cell by activation of the unfolded protein response and upregulation of CHOP. *Cell Death Dis* **3**, e322, doi:10.1038/cddis.2012.66 (2012).
- 8 Yue, X., Lio, C. J., Samaniego-Castruita, D., Li, X. & Rao, A. Loss of TET2 and TET3 in regulatory T cells unleashes effector function. *Nature communications* **10**, 2011, doi:10.1038/s41467-019-09541-y (2019).
- 9 Stefan-Lifshitz, M. *et al.* Epigenetic modulation of β cells by interferon- α via PNPT1/mir-26a/TET2 triggers autoimmune diabetes. *JCI Insight* **4**, doi:10.1172/jci.insight.126663 (2019).
- 10 Xu, Y. P. *et al.* Tumor suppressor TET2 promotes cancer immunity and immunotherapy efficacy. *The Journal of clinical investigation* **130**, 4316-4331, doi:10.1172/JCI129317 (2019).
- 11 Cardozo, A. K., Kruhoffer, M., Leeman, R., Orntoft, T. & Eizirik, D. L. Identification of novel cytokine-induced genes in pancreatic β -cells by high-density oligonucleotide arrays. *Diabetes* **50**, 909-920, doi:10.2337/diabetes.50.5.909 (2001).
- 12 Pirot, P., Cardozo, A. K. & Eizirik, D. L. Mediators and mechanisms of pancreatic β -cell death in type 1 diabetes. *Arq Bras Endocrinol Metabol* **52**, 156-165, doi:10.1590/s0004-27302008000200003 (2008).
- 13 Donath, M. Y., Boni-Schnetzler, M., Ellingsgaard, H., Halban, P. A. & Ehses, J. A. Cytokine production by islets in health and diabetes: cellular origin, regulation and function. *Trends in endocrinology and metabolism: TEM* **21**, 261-267, doi:10.1016/j.tem.2009.12.010 (2010).
- 14 Cardozo, A. K. *et al.* IL-1 β and IFN- γ induce the expression of diverse chemokines and IL-15 in human and rat pancreatic islet cells, and in islets from pre-diabetic NOD mice. *Diabetologia* **46**, 255-266, doi:10.1007/s00125-002-1017-0 (2003).

- 15 Antonelli, A., Ferrari, S. M., Corrado, A., Ferrannini, E. & Fallahi, P. CXCR3, CXCL10 and type 1 diabetes. *Cytokine Growth Factor Rev* **25**, 57-65, doi:10.1016/j.cytogfr.2014.01.006 (2014).

Reviewers' Comments:

Reviewer #1:

Remarks to the Author:

The Authors have answered in an adequate way most of my concerns.

There is, however, one remaining question. It is mentioned in the answers to the reviewers that Figure 2 describes experiments with 5 independent human islet donors, but this is not clearly explained in the legend for the Figure 2, which keeps mentioning one non-diabetic donor. Furthermore, a table should be provided describing characteristics of all human islet preparations used in the study (source, age, cause of death, purity of the preparations etc - see uniform request for human islet description in Diabetes or Diabetologia)

Reviewer #2:

Remarks to the Author:

The authors are to be commended for their careful responses to the reviewers' concerns. The revised manuscript has addressed most of the reviewers comments, with some issues beyond the scope of the current study.

Reviewer #3:

Remarks to the Author:

Thanks to the authors for their efforts towards the concerns raised. Overall the responses were satisfactory albeit not overwhelmingly convincing.

Remaining issues/comments:

The final sentences of the abstract and some of the main text imply possible implications for regulation of Tet2 activity towards therapeutic benefits. This idea is one of the attractive aspects of the work. Data in heterozygote mice are the most relevant in these contexts. Were all these mice and data thrown away? In order to generate proper littermate controls for the WT and KO conditions, ~50% of the mice must have been heterozygotes. Addition of this subgroup across the key experiments of the paper would provide important support for the authors' conclusions as opposed to one in that should be reworded to highlight that the data indicate that lifelong lack of TET2 is sufficient to disrupt diabetes triggering, but not infiltration, in NOD-diabetes.

Original comment ~157: The authors point to Supplemental Fig 1c-1e. Figure 1e appears to be missing data and contains evidence of some variation that potentially originates from the housekeeping gene quantification. Three data points are shown where 4 sorts of 4-5 mice were performed. Please explain the discrepancy (the latter) and add a second housekeeping gene. It is intriguing how perfectly aligned the KO and WT results are despite such a low N. With all due respect, I would recommend that the raw data and calculations be carefully re-examined by all individuals involved.

Reviewer #4:

Remarks to the Author:

The manuscript is very interesting. Previous concerns have been fully addressed. No more other comments.

REVIEWER COMMENTS

Reviewer #1 (Remarks to the Author):

The Authors have answered in an adequate way most of my concerns.

There is, however, one remaining question. It is mentioned in the answers to the reviewers that Figure 2 describes experiments with 5 independent human islet donors, but this is not clearly explained in the legend for the Figure 2, which keeps mentioning one non-diabetic donor. Furthermore, a table should be provided describing characteristics of all human islet preparations used in the study (source, age, cause of death, purity of the preparations etc - see uniform request for human islet description in Diabetes or Diabetologia)

We appreciate the Reviewer's supportive comments.

We apologize for the confusion and reworded the legend to figure 2 to "Data (mean \pm SEM) are from 5 experiments, each with 2,000-8,000 islet equivalency (IEQ) from non-diabetic individuals".

As suggested, following the guidelines in Diabetes, we added a table describing characteristics of all human islet preparations used in the study. Please see new Supplementary Table 1.

Reviewer #2 (Remarks to the Author):

The authors are to be commended for their careful responses to the reviewers' concerns. The revised manuscript has addressed most of the reviewers comments, with some issues beyond the scope of the current study.

We thank the Reviewer for the comments.

Reviewer #3 (Remarks to the Author):

Thanks to the authors for their efforts towards the concerns raised. Overall the responses were satisfactory albeit not overwhelmingly convincing.

We appreciate the Reviewer's supportive comments.

Remaining issues/comments:

The final sentences of the abstract and some of the main text imply possible implications for regulation of Tet2 activity towards therapeutic benefits. This idea is one of the attractive aspects of the work. Data in heterozygote mice are the most relevant in these contexts. Were all these mice and data thrown away? In order to generate proper littermate controls for the WT and KO conditions, ~50% of the mice must have been heterozygotes. Addition of this subgroup across the key experiments of the paper would provide important support for the authors' conclusions as opposed to one in that should be reworded to highlight that the data indicate that lifelong lack of TET2 is sufficient to disrupt diabetes triggering, but not infiltration, in NOD-diabetes.

The Tet2^{+/-} mice did not show the protection from spontaneous diabetes resistance to diabetes. This data is included in Supplemental Figure 2. For this reason they were not used for further studies with bone marrow transplantation.

We agree with the Reviewer's point that lack of Tet2 is sufficient to disrupt diabetes but not infiltration in NOD mice. However, we do not know the timing that is needed for protection and are planning future studies in which we will genetically eliminate Tet2 at specific times during the life of the mice.

We have modified the language in the abstract and text (in highlight) to reflect this limitation. We have added the following to the limitations section of the Discussion: In our *Tet2*-deficient NOD mice and recipients of bone marrow transplants, *Tet2* was absent throughout the development of immune cells. We do not know if there is a critical time(s) when *Tet2* expression is involved in β cell killing as well as immune activation but our adoptive transfer studies with diabetogenic T cells would suggest that absence of *Tet2* can be protective even when β cells are attacked by fully matured effector cells.

Original comment ~157: The authors point to Supplemental Fig 1c-1e. Figure 1e appears to be missing data and contains evidence of some variation that potentially originates from the housekeeping gene quantification. Three data points are shown where 4 sorts of 4-5 mice were performed. Please explain the discrepancy (the latter) and add a second housekeeping gene. It is intriguing how perfectly aligned the KO and WT results are despite such a low N. With all due respect, I would recommend that the raw data and calculations be carefully re-examined by all individuals involved.

We apologize for the confusion about the N's in the experiments and appreciate the Reviewer pointing out the problem. We have modified the legend to reflect the data that is shown: “Data are from 3 individual experiments, representing 4 sorts, n=4-5 mice each group per sort “

The N's for the experiments shown in this figure were quite robust: there were 16-20 mice used in both WT and KO group. The cells were perm/fixed and stained intracellularly for insulin to purify β cells by flow sorting. The RNA recovery rate from perm/fixed cells are usually extremely low and a large number of cells are needed to recover enough RNA for qRT-PCR. If the cell number was limiting, we pooled the cells from 2 sorts to have enough cells for one experiment. We have re-reviewed the data and can confirm that it is correct.

We also added the following description in the methods” In Supplementary Fig 1e, RNA was recovered from perm/fixed cells with RecoverAll™ Total Nucleic Acid Isolation Kit for FFPE (Invitrogen, AM1975). Taqman probes were used in qPCR as previously described 15.”

Reviewer #4 (Remarks to the Author):

The manuscript is very interesting. Previous concerns have been fully addressed. No more other comments.

We thank the Reviewer for the supportive comments.